# A Tailored Framework for Aligning Diffusion Models with Human Preference

## Abstract

Direct preference optimization (DPO) has shown success in aligning diffusion models with human preference. Previous approaches typically assume a consistent preference label between final generations and their corresponding noisy samples at intermediate steps, and directly apply DPO to these noisy samples for fine-tuning. However, we identify a significant issue with this assumption, as applying DPO to noisy samples from different trajectories based on final preferences may disrupt the optimization process. We first demonstrate inherent issues in previous methods from two perspectives: *gradient direction* and *preference order*, and then propose a **Tailor**ed **P**reference **O**ptimization (TailorPO) framework for aligning diffusion models with human preference, underpinned by some theoretical insights. Our approach directly ranks the preference of noisy samples based on their step-wise reward, and effectively resolves the gradient direction issues through a simple yet efficient design. Additionally, to the best of our knowledge, we are the first to consider the distinct structure of diffusion models and leverage the gradient guidance in preference aligning to enhance the optimization effectiveness. Experimental results demonstrate that our method significantly improves the model's ability to generate aesthetically pleasing and human-preferred images.

## 1 Introduction

Direct preference optimization (DPO), which fine-tunes models on paired data to align model outputs with human preferences, has demonstrated success in large language models (LLMs) (Rafailov et al., 2023). Recently, researchers generalized this method to diffusion models for text-to-image generation (Black et al., 2024; Yang et al., 2024a; Wallace et al., 2024). Given paired images generated from the same prompt and human-labeled preference, DPO increases the probability of generating the preferred sample while decreasing the probability of another sample, which enables the model to generate more visually appealing images that better align with human preferences.

Specifically, previous researchers (Yang et al., 2024a) leverage the *trajectory-level* preference to rank the generated samples. As shown in Figure 1(a), given a text prompt $c$, they first sample a pair of denoising trajectories $[x_T^0, \ldots, x_0^0]$ and $[x_T^1, \ldots, x_0^1]$ from the diffusion model, and then rank them according to the human preference on the final generated images $x_0^0$ and $x_0^1$. It is assumed that *the preference order of $(x_0^0, x_0^1)$, at the end of the generation trajectory, can consistently represent the preference order of $(x_t^0, x_t^1)$ at all intermediate steps $t$*. Then, the DPO loss function is implemented using the generation probabilities $p(x_{t-1}^0|x_t^0, c)$ and $p(x_{t-1}^1|x_t^1, c)$ at each step $t$ to fine-tune the diffusion model, which is called the *step-level* optimization.

However, we notice that the above trajectory-level preference ranking and the step-level optimization are not compatible in diffusion models. **First**, the trajectory-level preference ranking (*i.e.,* the preference order of final outputs $(x_0^0, x_0^1)$) does not accurately reflect the preference order of $(x_t^0, x_t^1)$ at intermediate steps. Considering the inherent randomness in the denoising process, simply assigning the preference of final outputs to all intermediate steps will detrimentally affect the preference optimization performance. **Second**, the generation probabilities $p(x_{t-1}^0|x_t^0, c)$ and $p(x_{t-1}^1|x_t^1, c)$ in two trajectories are conditioned on different inputs, and this causes the optimization direction to be significantly affected by input differences. In particular, if $x_t^0$ and $x_t^1$ are located in the same linear subspace of the diffusion model, then the optimization of DPO probably increases the probability of generating dis-preferred samples. Section 3.2 provides a detailed theoretical analysis of these issues.

Figure 1: Framework overview of (a) previous method and (b) TailorPO. In the previous method, the preference order is determined based on final outputs and used to guide the optimization of intermediate noisy samples in different generation trajectories. In contrast, we generate noisy samples from the same input $x_t$ and directly rank their preference order for optimization.

Therefore, in this paper, we propose a **Tailor**ed **P**reference **O**ptimization (TailorPO) framework to fine-tune diffusion models with DPO, which addresses the aforementioned challenges. As Figure 1(b) shows, we generate two noisy samples $(x_{t-1}^0, x_{t-1}^1)$ from the same input $x_t$ at each step. Then, we directly rank the preference order of two samples $(x_{t-1}^0, x_{t-1}^1)$ based on their step-wise reward. To this end, one of the most straightforward methods is to directly evaluate the reward of these noisy samples using a reward model. However, existing reward models are trained on natural images and do not apply to noisy samples. To address this issue, we formulate the denoising process as a Markov decision process (MDP) and derive a simple yet effective measurement for the reward of noisy samples. Then, we utilize $p(x_{t-1}^0|x_t, c)$ and $p(x_{t-1}^1|x_t, c)$ to compute the loss function for fine-tuning. In this way, the gradient direction is proven to increase the probability of generating preferred samples while decreasing the probability of generating dis-preferred samples.

Moreover, we notice that TailorPO generates paired samples from the same $x_t$, potentially causing two samples to be similar in late denoising steps with large $t$. Such similarity may reduce the diversity of paired samples, thereby impacting the effectiveness of the DPO-based method. To mitigate this issue, we propose to enhance the diversity of noisy samples by increasing their reward gap. Specifically, we employ gradient guidance (Guo et al., 2024) to generate paired samples, leveraging the gradient of differentiable reward models to increase the reward of preferred noisy samples. This strategy, termed *TailorPO-G*, further improves the effectiveness of our TailorPO framework.

In experiments, we fine-tune Stable Diffusion v1.5 using TailorPO and TailorPO-G to enhance its ability to generate images that achieve elevated aesthetic scores and align with human preference. Additionally, we evaluate TailorPO on user-specific preferences, such as image compressibility. The experimental results indicate that diffusion models fine-tuned with TailorPO and TailorPO-G yield higher reward scores compared to those fine-tuned with other RLHF and DPO-style methods.

Most close to our work, Liang et al. (2024) also noticed the inconsistency of the preference order between intermediate-step outputs and final images, and they proposed to train an additional step-wise reward model to address this issue. In comparison, we are the first to explicitly derive the theoretical flaws of previous DPO implementations in diffusion models, and we propose distinct solutions to address these issues. Experiments also demonstrate that our framework outperforms SPO on various reward models.

**Contributions** of this paper can be summarized as follows. (1) Through theoretical analysis and experimental validation, we demonstrate the mismatch between the trajectory-level ranking and the step-level optimization in existing DPO methods for diffusion models. (2) Based on these insights, we propose TailorPO, a simple DPO framework tailored to the unique denoising structure of diffusion models. To the best of our knowledge, this is the first framework that explicitly considers the properties of diffusion models for DPO. Experimental results have demonstrated that TailorPO significantly improves the model's ability to generate human-preferred images. (3) Furthermore, inspired by the success of gradient guidance in adapting model outputs towards user-specified objectives, we incorporate gradient guidance of differentiable reward models in TailorPO-G to increase the diversity of training samples for fine-tuning to further enhance performance.

## 2 RELATED WORKS

**Diffusion models.** As a new class of generative models, diffusion models (Sohl-Dickstein et al., 2015; Ho et al., 2020; Song et al., 2021) transform Gaussian noises into images (Dhariwal &

Nichol, 2021; Ho et al., 2022b; Nichol et al., 2022; Rombach et al., 2022), audios (Liu et al., 2023), videos (Ho et al., 2022a; Singer et al., 2023), 3D shapes (Zeng et al., 2022; Poole et al., 2023; Gu et al., 2023), and robotic trajectories (Janner et al., 2022; Chen et al., 2024) through an iterative process. Dhariwal & Nichol (2021) and Ho & Salimans (2022) further propose the classifier guidance and classifier-free guidance to align images with specific descriptions for text-to-image synthesis.

**Learning diffusion models from human feedback.** Inspired by the success of reinforcement learning from human feedback (RLHF) in large language models (Ouyang et al., 2022; Bai et al., 2022; OpenAI, 2023), many reward models have been developed for images preference, including aesthetic scorer (Schuhmann et al., 2022), ImageReward (Xu et al., 2023), PickScore model (Kirstain et al., 2023), and HPSv2 (Wu et al., 2023). Based on these reward models, Lee et al. (2023), DPOK (Fan et al., 2023) and DDPO (Black et al., 2024) formulated the denoising process of diffusion models as MDP and fine-tuned diffusion models using the policy-gradient method. DRaFT (Clark et al., 2024) and AlignProp (Prabhudesai et al., 2023) directly back-propagated the gradient of reward models through the sampling process for fine-tuning. In comparison, D3PO Yang et al. (2024a) and Diffusion DPO (Wallace et al., 2024) adapted the direct preference optimization (DPO) (Rafailov et al., 2023) on diffusion models and optimized model parameters at each denoising step. Considering the sequential nature of the denoising process, DenseReward (Yang et al., 2024b) assigned larger weights for initial steps than later steps when using DPO. Most close to our work, SPO (Liang et al., 2024) also pointed out the problematic assumption about the preference consistency of intermediate noisy samples and final images. However, they addressed this by training a step-wise reward model on another uncertain assumption. In comparison, we conduct a detailed analysis of the assumption and develop a new framework to improve the performance of DPO.

## 3 METHOD

### 3.1 PRELIMINARIES

**Diffusion models.** Diffusion models contain a forward process and a reverse denoising process. In the forward process, given an input $x_0$ sampled from the real distribution $p_{\text{data}}$, diffusion models gradually add Gaussian noises to $x_0$ at each step $t \in [1, T]$, as follows:

$$x_t = \sqrt{\alpha_t} x_{t-1} + \sqrt{1 - \alpha_t} \epsilon_{t-1} = \sqrt{\bar{\alpha}_t} x_0 + \sqrt{1 - \bar{\alpha}_t} \epsilon \tag{1}$$

where $\epsilon_t \sim \mathcal{N}(\mathbf{0}, \mathbf{I})$ denotes the Gaussian noise at step $t$. $\alpha_{1:T}$ denotes the variance schedule and $\bar{\alpha}_t = \prod_{i=1}^{t} \alpha_i$.

In the reverse denoising process, the diffusion model is trained to learn $p(x_{t-1}|x_t)$ at each step $t$. Specifically, following (Song et al., 2021), the denoising step at step $t$ is formulated as

$$x_{t-1} = \sqrt{\bar{\alpha}_{t-1}} \underbrace{\left( \frac{x_t - \sqrt{1 - \bar{\alpha}_t} \epsilon_\theta(x_t, t)}{\sqrt{\bar{\alpha}_t}} \right)}_{\hat{x}_0(x_t), \text{ predicted } x_0} + \underbrace{\sqrt{1 - \bar{\alpha}_{t-1} - \sigma_t^2} \epsilon_\theta(x_t, t)}_{\text{direction pointing to } x_t} + \underbrace{\sigma_t \epsilon_t'}_{\text{random noise}} \tag{2}$$

where $\epsilon_\theta(\cdot)$ is a noise prediction network with trainable parameters $\theta$, which aims to use $\epsilon_\theta(x_t, t)$ to predict the noise $\epsilon$ in Eq. (1) at each step $t$. $\epsilon_t' \sim \mathcal{N}(\mathbf{0}, \mathbf{I})$ is sampled from the standard Gaussian distribution. In fact, $x_{t-1}$ is sampled from the estimated distribution $\mathcal{N}(\mu_\theta(x_t), \sigma_t^2 \mathbf{I})$. According to the reverse process, $\hat{x}_0(x_t) = (x_t - \sqrt{1 - \bar{\alpha}_t} \epsilon_\theta(x_t, t) / \sqrt{\bar{\alpha}_t}$ represents the predicted $x_0$ at step $x$.

**Direct preference optimization (DPO) (Rafailov et al., 2023).** The DPO method was originally proposed to fine-tune large language models to align with human preferences based on paired datasets. Given a prompt $x$, two responses $y_0$ and $y_1$ are sampling from the generative model $\pi_\theta$, i.e., $y_0, y_1 \sim \pi_\theta(y|x)$. Then, $y_0$ and $y_1$ are ranked based on human preferences or the outputs $r(x, y_0)$ and $r(x, y_1)$ of a pre-trained reward model $r(\cdot)$. Let $y_w$ denote the preferred response in $(y_0, y_1)$ and $y_l$ denote the dis-preferred response. DPO optimizes parameters $\theta$ in $\pi_\theta$ by minimizing the following loss function.

$$\mathcal{L}_{\text{DPO}}(\theta) = -\mathbb{E}_{(x, y_w, y_l)} \left[ \log \sigma \left( \beta \log \frac{\pi_\theta(y_w|x)}{\pi_{\text{ref}}(y_w|x)} - \beta \log \frac{\pi_\theta(y_l|x)}{\pi_{\text{ref}}(y_l|x)} \right) \right] \tag{3}$$

where $\sigma$ is the sigmoid function, and $\beta$ is a hyper-parameter. $\pi_{\text{ref}}$ represents the reference model, usually set as the pre-trained models before fine-tuning. The gradient of the above loss function on

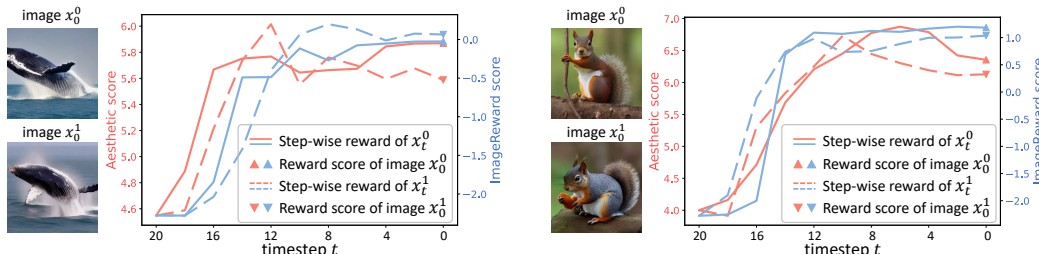

Figure 2: The preference order of intermediate noisy samples is not always consistent with the preference order of final output images, from both perspectives of the aesthetic score (red) and ImageReward score (blue).

each pair of $(x, y_w, y_l)$ with respect to the parameters $\theta$ is as follows (Rafailov et al., 2023).

$$\nabla_\theta \mathcal{L}_{\text{DPO}}(\theta, x, y_w, y_l) = -f(x, y_w, y_l) \left( \nabla_\theta \log \pi_\theta(y_w|x) - \nabla_\theta \log \pi_\theta(y_l|x) \right) \quad (4)$$

where $f(x, y_w, y_l) \triangleq \beta(1 - \sigma(\beta \log \frac{\pi_\theta(y_w|x)}{\pi_{\text{ref}}(y_w|x)} - \beta \log \frac{\pi_\theta(y_l|x)}{\pi_{\text{ref}}(y_l|x)}))$. Therefore, the gradient of the DPO loss function increases the likelihood of the preferred response $y_w$ and decreases the likelihood of the dis-preferred response $y_l$.

### 3.2 MISMATCH BETWEEN TRAJECTORY-LEVEL RANKING AND STEP-LEVEL OPTIMIZATION

In this section, we first revisit how existing works implement DPO for diffusion models, using D3PO (Yang et al., 2024a) as an example for explanation. Then, we identify the mismatch between their trajectory-level ranking and step-level optimization from two perspectives.

For a text-to-image diffusion model $\pi_\theta$ parameterized by $\theta$, given a text prompt $c$, D3PO first samples a pair of generation trajectories $[x_T^0, \ldots, x_0^0]$ and $[x_T^1, \ldots, x_0^1]$. Then, they compare the reward scores $r(c, x_0^0)$ and $r(c, x_0^1)$ of generated images, using the reward model $r(\cdot)$, and rank their preference order. The preferred image is denoted by $x_0^w$ and the dis-preferred image is denoted by $x_0^l$. Then, as Figure 1(a) shows, it is assumed that the preference order of final images $(x_0^0, x_0^1)$ represents the preference order of $(x_t^0, x_t^1)$ at all intermediate steps $t$. Subsequently, the diffusion model is fine-tuned by minimizing the following DPO-like loss function at the step level.

$$\mathcal{L}_{\text{D3PO}}(\theta) = -\mathbb{E}_{(c, x_t^w, x_t^l, x_{t-1}^w, x_{t-1}^l)} \left[ \log \sigma \left( \beta \log \frac{\pi_\theta(x_{t-1}^w|x_t^w, c)}{\pi_{\text{ref}}(x_{t-1}^w|x_t^w, c)} - \beta \log \frac{\pi_\theta(x_{t-1}^l|x_t^l, c)}{\pi_{\text{ref}}(x_{t-1}^l|x_t^l, c)} \right) \right] \quad (5)$$

We argue that there are two critical issues in the aforementioned process and loss function, which we will elaborate on and prove through the theoretical analysis in the following sections.

**Inaccurate preference order.** The first obvious issue is that the preference order of final images $x_0$ at the end of the trajectory cannot accurately reflect the preference order of noisy samples $x_t$ at intermediate steps. Liang et al. (2024) demonstrated that early steps in the denoising process tend to handle layout, while later steps focus more on detailed textures. However, the preference order based on final images primarily reflects layout and composition preferences, misaligning with the function of later steps. Taking a step further, we rethink this problem from another perspective and formulate the reward at intermediate steps based on theoretical analysis.

Similar to (Yang et al., 2024a), we formulate the denoising process as the following MDP.

$$S_t \triangleq (c, x_{T-t}), \ A_t \triangleq x_{T-t-1}, \ R_t = R(S_t, A_t) \triangleq R((c, x_{T-t}), x_{T-t-1})$$
$$P(S_{t+1}|S_t, A_t) \triangleq (\delta_c, \delta_{x_{T-t-1}}), \ \pi(A_t|S_t) \triangleq \pi_\theta(x_{T-t-1}|x_{T-t}, c) \quad (6)$$

By formulating the denoising process of the diffusion model as MDP, we aim to maximize the action value function at time $t$, *i.e.*, $Q(s, a) = \mathbb{E}[G_t|S_t = s, A_t = a]$, where $G_t$ represents the cumulative return at step $t$. We define $G_t$ in the general form of $TD(\lambda)$, $G_t^\lambda = (1 - \lambda) \sum_{n=1}^{T-t-1} \lambda^{n-1} G_t^{(n)} + \lambda^{T-t-1} G_t^{(T-t)}$, where $G_t^{(n)} = \sum_{i=1}^n \gamma^{i-1} R_{t+i} + \gamma^n V(S_{t+i})$ denotes the estimated return at step $t$ based on $n$ subsequent steps. Here, we simplify the analysis to $TD(1)$ and it degrades to the Monte Carlo method. In other words, we have $G_t^\lambda = G_t^{(T-t)}$. For diffusion models, we assume $R_t = 0$ for

Figure 3: Framework of TailorPO. At each step $t$, we start from the same $x_t$ to generate two noisy samples $x_{t-1}^0$ and $x_{t-1}^1$. Subsequently, we compare their step-wise reward to determine their preference order. For the preferred sample, if the reward model is differentiable, we employ the gradient guidance to further increase its reward to obtain $x_{t-1}^+$. Then, we optimize the generating probability of preferred and dis-preferred samples. After the optimization at step $t$, the preferred sample is taken as the input $x_{t-1}$ of the next step for later sampling and optimization.

$t < T$ to further simplify the return as $\gamma^n V(S_T)$. By replacing $\gamma = 1$ and $V(S_T) = R_T = r(c, x_0)$, which is the reward value of generated images, we have the following action value function.

$$Q(s, a) = \mathbb{E}[r(c, x_0)|S_t = (c, x_{T-t}), A_t = x_{T-t-1}] = \mathbb{E}[r(c, x_0)|c, x_{T-t-1}] \tag{7}$$

In other words, the quality of noisy samples $x_{T-t-1}$ can be determined by the expected reward value of images generated by different trajectories starting from $x_{T-t-1}$. In contrast, the reward value $r(c, x_0)$ of an image from a single trajectory does not represent the quality of the intermediate denoising action. Based on this analysis, we demonstrate that *the preference order of final images cannot accurately represent the preference order of intermediate noisy samples.*

To better illustrate this issue, we first propose a method for evaluating the quality of intermediate noisy samples, followed by an experimental validation using this method. The results shown in Figure 2 demonstrate that the preference order between a pair of intermediate samples $x_t$ can sometimes conflict with the preference order between the corresponding denoised images $x_0$. This finding likewise provides evidence against the validity of the assumption employed in previous methods. The proposed evaluation method and our framework will be elaborated in the subsequent sections.

**Disturbed gradient direction.** Moreover, even if we obtain an accurate preference order of noisy samples at intermediate steps, the loss function in Eq. (5) still has limitations from the gradient perspective. To gain a mechanistic understanding of the above loss function, we compute its gradient with respect to parameters $\theta$ as follows (please refer to Appendix A for the proof).

$$\nabla_\theta \mathcal{L}_{\text{D3PO}}(\theta) = -\mathbb{E}\left[(f_t/\sigma_t^2) \cdot [(x_{t-1}^w - \mu_\theta(x_t^w))^T \nabla_\theta \mu_\theta(x_t^w) - (x_{t-1}^l - \mu_\theta(x_t^l))^T \nabla_\theta \mu_\theta(x_t^l)]\right]$$

$$f_t \triangleq \beta(1 - \sigma(\beta \log \frac{\pi_\theta(x_{t-1}^w|x_t^w, c)}{\pi_{\text{ref}}(x_{t-1}^w|x_t^w, c)} - \beta \log \frac{\pi_\theta(x_{t-1}^l|x_t^l, c)}{\pi_{\text{ref}}(x_{t-1}^l|x_t^l, c)})) \tag{8}$$

In the above equation, the gradient is significantly affected by the relationship between inputs $x_t^w$ and $x_t^l$ from the previous step. This is because the input conditions $(x_t^w, x_t^l)$ of generation probabilities for preferred sample $x_{t-1}^w$ and dis-preferred sample $x_{t-1}^l$ in Eq. (5) are different. Therefore, the choice of $x_t^w$ and $x_t^l$ disturbs the original optimization direction of DPO. In particular, if $\nabla_\theta \mu_\theta(x_t^w) \approx \nabla_\theta \mu_\theta(x_t^l)$, then the gradient term can be written as:

$$\nabla_\theta \mathcal{L}_{\text{D3PO}}(\theta) \approx -\mathbb{E}\left[(f_t/\sigma_t^2) \cdot \nabla_\theta^T \mu_\theta(x_t^w)[(x_{t-1}^w - x_{t-1}^l) + (\mu_\theta(x_t^l) - \mu_\theta(x_t^w))]\right] \tag{9}$$

It shows that if $x_t^w$ and $x_t^l$ are located in the same linear subspace, then the optimization direction of the model shifts towards the direction $\mu_\theta(x_t^l) - \mu_\theta(x_t^w)$, which points to the dis-preferred samples. Thus, the fine-tuning effectiveness of DPO is significantly weakened.

In this section, we have conducted a detailed analysis of the denoising process based on MDP, and the optimization gradient of diffusion models. In this way, we reveal the potential theoretical issues in previous methods beyond visual discoveries of Liang et al. (2024). To address these issues, we propose distinct solutions in Section 3.3.

### 3.3 TAILORED PREFERENCE OPTIMIZATION FRAMEWORK FOR DIFFUSION MODELS

To address the aforementioned problems, considering the characteristics of diffusion models, we propose a **Tailor**ed **P**reference **O**ptimization (TailorPO) framework for fine-tuning diffusion models.

Specifically, given a text prompt $c$ and the timestep $t$, we always start from the **same** $x_t$ to generate the next time-step noisy samples, *i.e.,* $x_{t-1}^0$ and $x_{t-1}^1$. Then, we estimate the step-wise reward of intermediate noisy samples $x_{t-1}^0$ and $x_{t-1}^1$ to directly rank their preference order. The sample with the higher reward value is represented by $x_{t-1}^w$, and the sample with the lower reward is denoted as $x_{t-1}^l$. Furthermore, if the reward function is differentiable, we apply the gradient guidance of the reward function (introduced in Section 3.4) to increase the reward of the preferred sample $x_{t-1}^w$, which enlarges the reward gap between $x_{t-1}^w$ and $x_{t-1}^l$ and enhances the fine-tuning effectiveness. At the next denoising step $(t-1)$, the preferred sample $x_{t-1}^w$ is taken as $x_{t-1}$ for further sampling and training. Our framework is illustrated in Figure 3, and the loss function is given as follows.

$$\mathcal{L}(\theta) = -\mathbb{E}_{(c,x_t,x_{t-1}^w,x_{t-1}^l)} \left[ \log \sigma \left( \beta \log \frac{\pi_\theta(x_{t-1}^w|x_t,c)}{\pi_{\text{ref}}(x_{t-1}^w|x_t,c)} - \beta \log \frac{\pi_\theta(x_{t-1}^l|x_t,c)}{\pi_{\text{ref}}(x_{t-1}^l|x_t,c)} \right) \right] \tag{10}$$

We will subsequently elucidate and substantiate the advantages of our proposed TailorPO framework for diffusion models from the following perspectives.

**Consistency between gradient direction and preferred samples.** First, TailorPO addresses the problem with the gradient direction of previous methods by always generating paired samples from the same $x_t$. Different from (Liang et al., 2024), we aim to address the gradient issue in Section 3.2 and it is straightforward to sample from the same $x_t$ based on our theoretical analysis. This simple operation ensures that the generation probabilities in Eq. (10) are all based on the same condition, aligning with the original formulation of DPO in Eq. (3). In this way, the gradient of our loss function is given as follows (please refer to Appendix A for the proof).

$$\nabla_\theta \mathcal{L}(\theta) = -\mathbb{E} \left[ (f_t/\sigma_t^2) \cdot \nabla_\theta^T \mu_\theta(x_t)(x_{t-1}^w - x_{t-1}^l) \right] \tag{11}$$

Notably, the gradient direction of our loss function clearly points towards the preferred samples. Therefore, the model is effectively encouraged to generate preferred samples.

**Immediate preference ranking at intermediate steps.** Instead of performing preference ranking on final images, we directly rank the preference order of noisy samples at intermediate steps. To this end, Liang et al. (2024) proposed to train a step-wise reward model based on another uncertified assumption, *i.e.,* "the preference order between pair of images can be kept when adding the same noise." In comparison, we directly evaluate the preference quality of noisy samples $x_t$ without training a new model. As discussed in Section 3.2, the denoising process can be formulated as an MDP, where the action value function for generating $x_t$ simplifies to the expected reward of images over all trajectories starting from $x_t$. Therefore, we define the step-wise reward value of the noisy sample $x_t$ as follows.

$$r_t(c,x_t) \triangleq \mathbb{E}[r(c,x_0)|c,x_t] \approx r(c,\hat{x}_0(x_t)) \tag{12}$$

However, computing the above expectation over all trajectories is intractable. Therefore, we employ an approximation to the expectation value. Previous studies (Chung et al., 2023; Guo et al., 2024) have proven that $\mathbb{E}[x_0|c,x_t] = \hat{x}_0(x_t)$, which represents the predicted $x_0$ at step $t$ (defined in Eq. (2)). Furthermore, Chung et al. (2023) prove the following Proposition 1, which ensures that the expectation of image rewards $\mathbb{E}[r(c,x_0)|c,x_t]$ can be approximated by the reward of the expected image $r(c,\mathbb{E}[x_0|c,x_t])$. Therefore, we compute $r_t(c,x_t) \approx r(c,\hat{x}_0(x_t))$ to estimate the step-wise reward of $x_t$ for preference ranking.

**Proposition 1 (proven by Chung et al. (2023))** *Let a measurement $g(x_0) = \mathcal{A}(x_0) + n$, where $\mathcal{A}(\cdot)$ is a measure operator defined on images $x_0$ and $n \sim \mathcal{N}(0, \sigma^2 I)$ is the measurement noise. The Jensen gap between $\mathbb{E}[g(x_0)|c,x_t]$ and $g(\mathbb{E}[x_0|c,x_t])$, i.e., $\mathcal{J} = \mathbb{E}[g(x_0)|c,x_t] - g(\mathbb{E}[x_0|c,x_t])$ is bounded by $\mathcal{J} \leq \frac{d}{\sqrt{2\pi\sigma^2}} e^{-1/2\sigma^2} \|\nabla_x \mathcal{A}(x)\| m_1$, where $\nabla_x \mathcal{A}(x) \triangleq \max_x \|\nabla_x \mathcal{A}(x)\|$, $m_1 \triangleq \int \|x_0 - \hat{x}_0\| p(x_0|c,x_t) dx_0$, and $\hat{x}_0 = \mathbb{E}[x_0|c,x_t]$. The Jensen gap can approach 0 as $\sigma$ increases.*

By obtaining the preference order of noisy samples immediately at intermediate steps, we can fine-tune the model using Eq. (10). Then, the preferred sample $x_{t-1}^w$ is assigned as the input for the next step, enabling sampling and optimization in subsequent steps.

### 3.4 GRADIENT GUIDANCE OF REWARD MODEL FOR FINE-TUNING

In TailorPO, since noisy samples $(x_{t-1}^0, x_{t-1}^1)$ are generated from the same $x_t$, their similarity increases as $t$ decreases. This increasing similarity potentially reduces the diversity of paired samples

Table 1: Gradient guidance successfully increased/decreased the reward of most samples.

| $t$ | 20 | 16 | 12 | 8 | 4 |
|---|---|---|---|---|---|
| ratio of $r_t(c, x_{t-1}^+) > r_t(c, x_{t-1})$ | 0.83 | 0.97 | 0.98 | 0.99 | 0.99 |
| ratio of $r_t(c, x_{t-1}^-) < r_t(c, x_{t-1})$ | 0.87 | 0.98 | 1.00 | 0.98 | 1.00 |

---

**Algorithm 1:** The TailorPO-G framework for aligning diffusion models with human preference.

**Input:** Diffusion model $\pi_\theta(\cdot)$, reference model $\pi_{\text{ref}}(\cdot)$, reward model $r(\cdot)$

1 Sample a text prompt $c$;
2 Initialize $x_T \sim \mathcal{N}(0, \boldsymbol{I})$;
3 **for** $t = T, \ldots, 1$ **do**
4      Sample $x_{t-1}^0, x_{t-1}^1$ from $\pi_\theta(\cdot|x_t, c)$;
5      Rank $x_{t-1}^0$ and $x_{t-1}^1$ based on their step-wise rewards to obtain $x_{t-1}^w$ and $x_{t-1}^l$;
6      Inject gradient guidance to compute $x_{t-1}^+ = x_{t-1}^w - \eta_t \nabla_{x_{t-1}^w}(r_{\text{high}} - r_t(c, x_{t-1}^w))^2$;
7      **if** $r_t(c, x_{t-1}^+) > r_t(c, x_{t-1}^w)$ **then**
8          $x_{t-1}^w \leftarrow x_{t-1}^+$
9      **end**
10      Optimize $\pi_\theta(\cdot)$ using Eq. (10);
11      $x_{t-1} \leftarrow x_{t-1}^w$;
12 **end**

**Output:** The fine-tuned diffusion model $\pi_\theta(\cdot)$.

---

for training. On the other hand, Khaki et al. (2024) have shown that a large difference between paired samples is beneficial to the DPO effectiveness. Therefore, to enhance the DPO performance in this case, we propose enlarging the difference between two noisy samples from the reward perspective.

To this end, we consider how to adjust the reward of a noisy sample $x_{t-1}$. Similar to (Guo et al., 2024), we use $r_{\text{high}}$ to represent an expected higher reward. Then, the gradient of the conditional score function is $\nabla_{x_{t-1}} \log p(x_{t-1}|r_{\text{high}}) = \nabla \log p(x_{t-1}) + \nabla_{x_{t-1}} \log p(r_{\text{high}}|x_{t-1})$, where the first term $\nabla \log p(x_{t-1})$ is estimated by the diffusion model itself, and the second term is to be estimated by the guidance. Guo et al. (2024) further prove the following relationship for estimation.

$$\nabla_{x_{t-1}} \log p(r_{\text{high}}|x_{t-1}) \propto \nabla_{x_{t-1}} \log p(r_{\text{high}}|\hat{x}_0(x_{t-1})) \propto -\eta_t \nabla_{x_{t-1}} (r_{\text{high}} - r_t(c, x_{t-1}))^2 \quad (13)$$

Therefore, we can inject the gradient term $\nabla_{x_{t-1}}(r_{\text{high}} - r_t(c, x_{t-1}))^2$ as the guidance to the generation of $x_{t-1}$ to adjust its reward. Specifically, we update the noisy samples as follows.

$$
\begin{aligned}
x_{t-1}^+ &\leftarrow x_{t-1} - \eta_t \nabla_{x_{t-1}}(r_{\text{high}} - r_t(c, x_{t-1}))^2, \text{ to increase reward} \\
x_{t-1}^- &\leftarrow x_{t-1} + \eta_t \nabla_{x_{t-1}}(r_{\text{high}} - r_t(c, x_{t-1}))^2, \text{ to decrease reward}
\end{aligned}
\quad (14)
$$

To demonstrate that the above gradient guidance is able to adjust the reward of noisy samples as expected, we compared the step-wise rewards of the original sample $x_{t-1}$, the increased sample $x_{t-1}^+$, and the decreased sample $x_{t-1}^-$. Specifically, we generated 100 noisy samples $x_{t-1}$ from Stable Diffusion v1.5 (Rombach et al., 2022), and then computed the corresponding $x_{t-1}^+$ and $x_{t-1}^-$. We set $\eta_t = 0.2$ and $r_{\text{high}} = r_t(c, x_{t-1}) + \delta$ following Guo et al. (2024), where the constant $\delta = 0.5$ specified the expected increment of the reward value.

Then, we computed the ratio of increased samples (satisfying $r_t(c, x_{t-1}^+) > r_t(c, x_{t-1})$) and the ratio of decreased samples (satisfying $r_t(c, x_{t-1}^-) < r_t(c, x_{t-1})$). Table 1 shows that for almost all samples, the gradient guidance successfully increased or decreased their reward as expected, demonstrating its effectiveness in adapting the reward of samples.

Finally, we apply this method in our training process to enlarge the reward gap between a pair of noisy samples and develop the *TailorPO-G* framework. As shown in Figure 3 and Algorithm 1, we first modify the preferred sample $x_{t-1}^w$ to increase its reward value, and then use the modified sample for fine-tuning and subsequent sampling.

## 4 EXPERIMENTS

**Experimental settings.** In our experiments, we evaluate the effectiveness of our method in fine-tuning Stable Diffusion v1.5 (Rombach et al., 2022). We compared our TailorPO method with the

Table 2: Reward values of images generated by diffusion models fine-tuned using different methods. The prompts are related to common animals.

| | Aesthetic scorer | ImageReward | HPSv2 | PickScore | Compressibility |
|---|---|---|---|---|---|
| Stable Diffusion v1.5 | 5.79 | 0.65 | 27.51 | 20.20 | -105.51 |
| DDPO (Black et al., 2024) | 6.57 | 0.99 | 28.00 | 20.24 | -37.37 |
| D3PO (Yang et al., 2024a) | 6.46 | 0.95 | 27.80 | 20.40 | -29.31 |
| SPO (Liang et al., 2024) | 5.89 | 0.95 | 27.88 | 20.38 | – |
| TailorPO | 6.66 | 1.20 | **28.37** | 20.34 | **-6.71** |
| TailorPO-G | **6.96** | **1.26** | 28.03 | **20.68** | – |

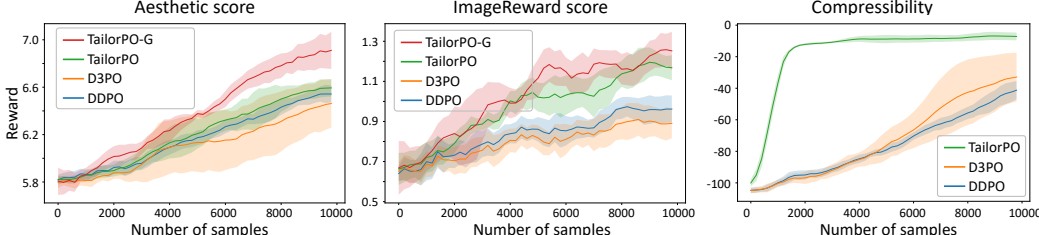

Figure 4: The change curve of reward values during the fine-tuning process. Experiments were conducted for three runs and we plot the average value and standard deviation of the reward.

RLHF method, DDPO (Black et al., 2024), and DPO-style methods, including D3PO (Yang et al., 2024a) and SPO (Liang et al., 2024). For all methods, we used the aesthetic scorer (Schuhmann et al., 2022), ImageReward (Xu et al., 2023), PickScore (Kirstain et al., 2023), HPSv2 (Wu et al., 2023), and JPEG compressibility measurement (Black et al., 2024) as reward models. Considering that some reward models are non-differentiable, we evaluate both the effectiveness of TailorPO and TailorPO-G, respectively.

Following the settings in D3PO (Yang et al., 2024a) and SPO (Liang et al., 2024), we applied the DDIM scheduler (Song et al., 2021) with $\eta = 1.0$ and $T = 20$ inference steps. The generated images were of resolution of $512 \times 512$. We employed LoRA (Hu et al., 2022) to fine-tune the UNet parameters on a total of 10,000 samples with a batch size of 2. The reference model was set as the pre-trained Stable Diffusion v1.5 itself. For SPO, we ran the officially released code by using the same hyper-parameters as in its original paper, and for other methods, we used the same hyper-parameters as in (Yang et al., 2024a), except that we set a smaller batch size for all methods. In particular, for all our frameworks, we generated images with $T = 20$ and uniformly sampled $T_{\text{fine-tune}} = 5$ steps for fine-tuning, $i.e.,$ we only fine-tuned the model at steps $t = 20, 16, 12, 8, 4$. In addition, we set the coefficient $\eta_t$ in gradient guidance using a cosine scheduler in the range of $[0.1, 0.2]$, which assigned a higher coefficient to smaller $t$ (samples closer to output images). We have conducted ablation studies in Appendix F to show that our method is relatively stable with respect to the setting of $T_{\text{fine-tune}}$ and $\eta_t$.

### 4.1 EFFECTIVENESS OF ALIGNING DIFFUSION MODELS WITH PREFERENCE

In this section, we demonstrate that our frameworks outperform previous methods in aligning diffusion models with various preferences, from both quantitative and qualitative perspectives.

**Quantitative evaluation.** We fine-tuned SD v1.5 on various reward models using a set of prompts of common animals released by Black et al. (2024) and a set of complex prompts in the Pick-a-Pic dataset (Kirstain et al., 2023), respectively. For quantitative evaluation, we randomly sampled five images for each prompt and computed the average reward value of all images. For the animal-related prompts, Table 2 demonstrates that both TailorPO and TailorPO-G outperform other methods across all reward models. On the other hand, Figure 4 shows curves of reward values throughout the fine-tuning process. It can be observed that our methods rapidly increase the reward of generations in early iterations. Appendix E compares results on prompts in the Pick-a-Pic dataset and shows that our method also effectively improved the reward values, surpassing SPO and the state-of-the-art offline method, Diffusion-DPO (Wallace et al., 2024). Notably, our methods outperform SPO as we directly estimate the step-level reward without training another reward model based on an uncertified assumption, and we incorporate the gradient guidance to further improve the effectiveness.

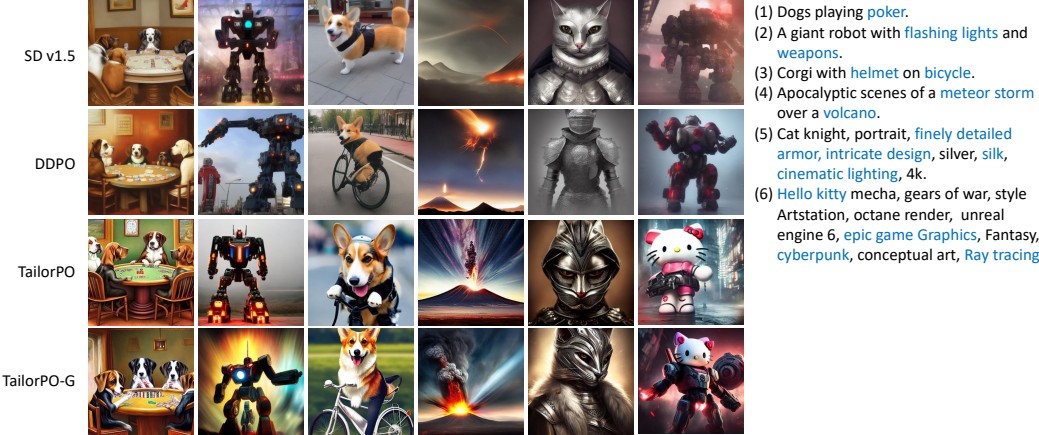

Figure 5: Visualization of images generated by diffusion models fine-tuned using different methods. For these animal-related prompts, diffusion models fine-tuned by TailorPO and TailorPO-G generated more colorful and visually pleasing images.

Figure 6: Visualization of images generated by diffusion models fine-tuned on complex prompts in the Pick-a-Pic dataset. Prompts are given on the right with missing elements in SD v1.5 highlighted.

(1) Dogs playing poker.
(2) A giant robot with flashing lights and weapons.
(3) Corgi with helmet on bicycle.
(4) Apocalyptic scenes of a meteor storm over a volcano.
(5) Cat knight, portrait, finely detailed armor, intricate design, silver, silk, cinematic lighting, 4k.
(6) Hello kitty mecha, gears of war, style Artstation, octane render,  unreal engine 6, epic game Graphics, Fantasy, cyberpunk, conceptual art, Ray tracing.

**Qualitative comparison.** For qualitative comparison, we first visualize the generated samples given simple animal prompts in Figure 5. It is obvious that after fine-tuning using TailorPO and TailorPO-G, the model generated more colorful and visually appealing images with fine-grained details. In addition, we fine-tuned SD v1.5 on more complex prompts, using 4k prompts in the Pick-a-Pic training dataset (Kirstain et al., 2023; Liang et al., 2024). Figure 6 shows that both TailorPO and TailorPO-G encourage the model to generate more aesthetically pleasing images, and these images were better aligned with the given prompts. For example, in the third row of Figure 6, the 5th and 6th images contained more consistent and aligned subjects, scenes, and elements with prompts.

**User study.** Additionally, we conducted a user study by requesting ten users to label their preference for generated images from the perspective of visual appeal and general preference. For each fine-tuned model, we generated images for each animal-related prompt and asked users to compare and annotate images generated by different models to indicate their preferences. Figure 7 reports the win-lose percentage results of our method versus other baseline methods, where our method exhibits a clear advantage in aligning with human preference. More experimental details and the ethics statement about the user study can be seen in Appendix D.

### 4.2 GENERALIZATION TO DIFFERENT PROMPTS AND REWARD MODELS

In this section, we investigate two types of generalization abilities of the fine-tuned model using our method, including prompt generalization and reward generalization (Clark et al., 2024).

**Prompt generalization** refers to the model's ability to generate high-quality images for prompts beyond those used in fine-tuning. To evaluate this, we fine-tuned Stable Diffusion v1.5 on 45

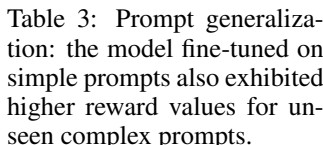



Figure 7: User-labeled win-lose ratio of TailorPO and TailorPO-G versus other baseline methods.

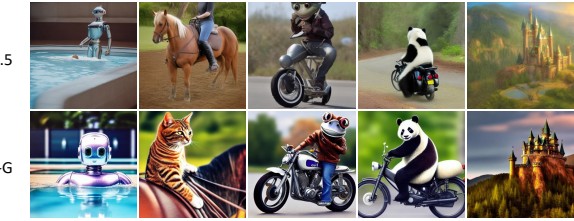

Figure 8: Diffusion model fine-tuned on simple prompts generalized well to complex prompts. Prompts from left to right are: (1) cinematic still of a stainless steel robot swimming in a pool. (2) A cat that is riding a horse without a leg. (3) Crazy frog, on one wheel, motorcycle, dead. (4) A panda riding a motorcycle. (5) Fantasy castle on a hilltop.

Table 3: Prompt generalization: the model fine-tuned on simple prompts also exhibited higher reward values for unseen complex prompts.

|  | Aesthetic scorer | ImageReward | HPSv2 | PickScore | Compressibility |
|---|---|---|---|---|---|
| SD v1.5 | 5.69 | -0.04 | 25.79 | 17.74 | -98.95 |
| DDPO | 5.94 | 0.06 | 26.24 | 17.74 | -49.94 |
| D3PO | 6.14 | 0.11 | 26.09 | 17.77 | -38.92 |
| SPO | 5.79 | 0.15 | 26.28 | 17.16 | – |
| TailorPO | 6.26 | 0.11 | **26.64** | 17.85 | **-7.32** |
| TailorPO-G | **6.45** | **0.25** | 26.25 | **17.93** | – |

Table 4: Reward generalization: the model fine-tuned towards a reward model also exhibited higher reward values on other different but related reward models.

| Evaluate
Train | Aesthetic scorer | ImageReward | HPSv2 | PickScore |
|---|---|---|---|---|
| SD v1.5 | 5.79 | 0.65 | 27.51 | 20.20 |
| Aesthetic scorer | **6.96** | 1.04 | 27.63 | 20.34 |
| ImageReward | 6.01 | **1.26** | 28.01 | 20.21 |
| HPSv2 | 5.45 | 0.92 | **28.03** | 20.04 |
| PickScore | 5.94 | 0.83 | 27.71 | **20.68** |

prompts of simple animal (Black et al., 2024) and evaluated its performance on 500 complex prompts (Kirstain et al., 2023). As shown in Table 3, the model fine-tuned on simple prompts exhibited higher reward values on complex prompts than the original SD v1.5, with our approach achieving the highest performance. Figure 8 presents examples of images generated from complex prompts, demonstrating that despite being fine-tuned on simple prompts, the model was also capable of generating high-quality images given complex prompts. This highlights the effectiveness of our method in enhancing the model's generalization to human-preferred images across various prompts, rather than overfitting to simple prompts.

**Reward generalization** refers to the phenomenon where fine-tuning the model towards a specific reward model can also enhance its performance on another different but related reward model. We selected one reward model from the aesthetic scorer, ImageReward, HPSv2, and Pickscore for fine-tuning, and used the other three reward models for evaluation. Table 4 shows that after being fine-tuned towards the aesthetic scorer, ImageReward, and PickScore, the model usually exhibited higher performance on all these four reward models. In other words, our method boosted the overall ability of the model to generate high-quality images.

## 5 CONCLUSIONS

In this study, we rethink the existing DPO framework for aligning diffusion models and identify the potential flaws in these methods. We analyze these issues from both perspectives of preference order and gradient direction. To address these challenges, we consider the unique characteristics of diffusion models and introduce a novel tailored preference optimization framework for aligning diffusion models with human preference. Specifically, at each denoising step, our approach generates noisy samples from the same input and directly ranks their preference order for optimization. Furthermore, we propose integrating gradient guidance into the training framework to enhance the training effectiveness. Experimental results demonstrate that our approach significantly improved the reward scores of generated images, and exhibited good generalization over different prompts and different reward models.

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

## A GRADIENT OF LOSS FUNCTIONS

**Gradient of the original DPO loss function.** Given the input $(x, y^w, y^l) \sim \mathcal{D}$, the loss of DPO is as follows.

$$\mathcal{L} = -\mathbb{E}_{(x, y_w, y_l) \sim \mathcal{D}}[\log \sigma(\beta \log \frac{\pi_\theta(y_w|x)}{\pi_{\text{ref}}(y_w|x)} - \beta \log \frac{\pi_\theta(y_l|x)}{\pi_{\text{ref}}(y_l|x)})] \tag{15}$$

Let $h_\theta(x, y_w, y_l) \triangleq \beta \log \frac{\pi_\theta(y_w|x)}{\pi_{\text{ref}}(y_w|x)} - \beta \log \frac{\pi_\theta(y_l|x)}{\pi_{\text{ref}}(y_l|x)}$ and $f(x, y_w, y_l) \triangleq \beta(1 - \sigma(h_\theta(x, y_w, y_l)))$, then

$$\begin{aligned}
\frac{\partial \mathcal{L}(x, y_w, y_l)}{\partial \theta} &= \frac{\partial - \log \sigma(h_\theta(x, y_w, y_l))}{\partial \theta} \\
&= -\frac{1}{\sigma(h_\theta(x, y_w, y_l))} \frac{\partial \sigma(h_\theta(x, y_w, y_l))}{\partial \theta} \\
&= -\frac{1}{\sigma(h_\theta(x, y_w, y_l))} \frac{\partial \sigma(h_\theta(x, y_w, y_l))}{\partial h_\theta(x, y_w, y_l)} \frac{\partial h_\theta(x, y_w, y_l)}{\partial \theta} \\
&= -\frac{1}{\sigma(h_\theta(x, y_w, y_l))} \sigma(h_\theta(x, y_w, y_l))(1 - \sigma(h_\theta(x, y_w, y_l))) \frac{\partial h_\theta(x, y_w, y_l)}{\partial \theta} \\
&= -f(x, y_w, y_l) \frac{\partial [\log \pi_\theta(y_w|x) - \log \pi_{\text{ref}}(y_w|x) - \log \pi_\theta(y_l|x) + \log \pi_{\text{ref}}(y_l|x)]}{\partial \theta} \\
&= -f(x, y_w, y_l)(\frac{\partial \log \pi_\theta(y_w|x)}{\partial \theta} - \frac{\partial \log \pi_\theta(y_l|x)}{\partial \theta})
\end{aligned} \tag{16}$$

**Gradient of the loss function of D3PO.** To study the generative distribution in the denoising process of diffusion models, let $x \triangleq (x_t, c), y \triangleq x_{t-1}$, then we have

$$\pi_\theta(y|x) = \pi_\theta(x_{t-1}|x_t, c) = \frac{1}{(2\pi\sigma_t^2)^{d/2}} \exp(-\frac{\|x_{t-1} - \mu_\theta(x_t)\|_2^2}{2\sigma_t^2}) \tag{17}$$

In this case, the gradient of the loglikelihood $\log \pi_\theta(x_{t-1}|x_t, c)$ *w.r.t.* $\theta$ is given as follows.

$$\begin{aligned}
\frac{\partial \log \pi_\theta(x_{t-1}|x_t, c)}{\partial \theta} &= (\frac{\partial \mu_\theta(x_t)}{\partial \theta})^T \frac{\partial(-\frac{\|x_{t-1} - \mu_\theta(x_t)\|_2^2}{2\sigma_t^2} - \log((2\pi\sigma_t^2)^{d/2}))}{\partial \mu_\theta(x_t)} \\
&= (\frac{\partial \mu_\theta(x_t)}{\partial \theta})^T \frac{(x_{t-1} - \mu_\theta(x_t))}{\sigma_t^2}
\end{aligned} \tag{18}$$

Then, we consider the gradient of the D3PO loss *w.r.t.* the model output $\mu_\theta$.

$$\begin{aligned}
\frac{\partial \mathcal{L}(x_t^w, x_{t-1}^w, x_t^l, x_{t-1}^l)}{\partial \theta} &= -f_t(\frac{\partial \log \pi_\theta(x_{t-1}^w|x_t^w, t, c)}{\partial \theta} - \frac{\partial \log \pi_\theta(x_{t-1}^l|x_t^l, t, c)}{\partial \theta}) \\
&= -\frac{f_t}{\sigma_t^2}\left[(\frac{\partial \mu_\theta(x_t^w)}{\partial \theta})^T(x_{t-1}^w - \mu_\theta(x_t^w)) - (\frac{\partial \mu_\theta(x_t^l)}{\partial \theta})^T(x_{t-1}^l - \mu_\theta(x_t^l))\right]
\end{aligned} \tag{19}$$

Suppose $\Delta\theta = -\eta\frac{\partial \mathcal{L}(x_t^w, x_{t-1}^w, x_t^l, x_{t-1}^l)}{\partial \theta}$. After the update of $\theta' \leftarrow \theta + \Delta\theta$, $\Delta\mu_\theta(x_t^w) \approx \eta\frac{f_t}{\sigma_t^2}[(\frac{\partial \mu_\theta(x_t^w)}{\partial \theta})(\frac{\partial \mu_\theta(x_t^w)}{\partial \theta})^T(x_{t-1}^w - \mu_\theta(x_t^w))] - \eta\frac{f_t}{\sigma_t^2}[(\frac{\partial \mu_\theta(x_t^w)}{\partial \theta})(\frac{\partial \mu_\theta(x_t^l)}{\partial \theta})^T(x_{t-1}^l - \mu_\theta(x_t^l))]$. If $x_t^w$ and $x_t^l$ are located in the same linear subspace of the model, *i.e.,* $\frac{\partial \mu_\theta(x_t^w)}{\partial \theta} \approx \frac{\partial \mu_\theta(x_t^l)}{\partial \theta}$, then the gradient can be written as follows.

$$\begin{aligned}
\frac{\partial \mathcal{L}(x_t^w, x_{t-1}^w, x_t^l, x_{t-1}^l)}{\partial \theta} &= -\frac{f_t}{\sigma_t^2}\left[(\frac{\partial \mu_\theta(x_t^w)}{\partial \theta})^T(x_{t-1}^w - \mu_\theta(x_t^w)) - (\frac{\partial \mu_\theta(x_t^l)}{\partial \theta})^T(x_{t-1}^l - \mu_\theta(x_t^l))\right] \\
&\approx -\frac{f_t}{\sigma_t^2}\left[(\frac{\partial \mu_\theta(x_t^w)}{\partial \theta})^T(x_{t-1}^w - \mu_\theta(x_t^w)) - (\frac{\partial \mu_\theta(x_t^w)}{\partial \theta})^T(x_{t-1}^l - \mu_\theta(x_t^l))\right] \\
&\approx -\frac{f_t}{\sigma_t^2}(\frac{\partial \mu_\theta(x_t^w)}{\partial \theta})^T\left[(x_{t-1}^w - x_{t-1}^l) + (\mu_\theta(x_t^l) - \mu_\theta(x_t^w))\right]
\end{aligned} \tag{20}$$

Suppose $\Delta\theta = -\eta\frac{\partial\mathcal{L}(x_t^w,x_{t-1}^w,x_t^l,x_{t-1}^l)}{\partial\theta}$. After the update of $\theta' \leftarrow \theta + \Delta\theta$, $\Delta\mu_\theta(x_t^w) \approx \eta\frac{f_t}{\sigma_t^2}(\frac{\partial\mu_\theta(x_t^w)}{\partial\theta})(\frac{\partial\mu_\theta(x_t^w)}{\partial\theta})^T[(x_{t-1}^w - x_{t-1}^l) + (\mu_\theta(x_t^l) - \mu_\theta(x_t^w))]$.

**Gradient of our loss function.** Then, we consider the gradient of our loss function *w.r.t.* the model output $\mu_\theta$.

$$
\begin{aligned}
\frac{\partial\mathcal{L}(x_t, x_{t-1}^w, x_{t-1}^l)}{\partial\theta} &= -f_t(\frac{\partial\mu_\theta(x_t)}{\partial\theta})^T(\frac{\partial\log\pi_\theta(x_{t-1}^w|x_t,t,c)}{\partial\mu_\theta(x_t)} - \frac{\partial\log\pi_\theta(x_{t-1}^l|x_t,t,c)}{\partial\mu_\theta(x_t)}) \\
&= -f_t(\frac{\partial\mu_\theta(x_t)}{\partial\theta})^T(\frac{x_{t-1}^w - \mu_\theta(x_t)}{\sigma_t^2} - \frac{x_{t-1}^l - \mu_\theta(x_t)}{\sigma_t^2}) \\
&= -\frac{f_t}{\sigma_t^2}(\frac{\partial\mu_\theta(x_t)}{\partial\theta})^T(x_{t-1}^w - x_{t-1}^l)
\end{aligned}
\tag{21}
$$

Suppose $\Delta\theta = -\eta\frac{\partial\mathcal{L}(x_t,x_{t-1}^w,x_{t-1}^l)}{\partial\theta}$. After the update of $\theta' \leftarrow \theta + \Delta\theta$, $\Delta\mu_\theta(x_t) \approx (\frac{\partial\mu_\theta(x_t)}{\partial\theta})\Delta\theta = \eta\frac{f_t}{\sigma_t^2}(\frac{\partial\mu_\theta(x_t)}{\partial\theta})(\frac{\partial\mu_\theta(x_t)}{\partial\theta})^T(x_{t-1}^w - x_{t-1}^l)$.

# B  TAILORPO AND TAILORPO-G

In this section, we provide another formulation for the loss function of TailorPO, and then discuss the difference between TailorPO and TailorPO-G from the perspective of gradient.

First, Eq. (10) only shows a classic loss formulation of DPO, and does not reflect the preference selection procedure in TailorPO. To this end, we provide another formulation of the loss function, which incorporates the preference selection based on step-wise reward $r_t$.

$$
\begin{aligned}
\mathcal{L}(\theta) &= -\mathbb{E}_{(c,x_t,x_{t-1}^{(0)},x_{t-1}^{(1)})}\left[\log\sigma\left((-1)^{\mathbb{1}(r_t(c,x_{-1}^{(0)})<r_t(c,x_{-1}^{(1)}))}\cdot\Delta\right)\right], \\
\Delta &= \beta\log\frac{\pi_\theta(x_{t-1}^{(0)}|x_t,c)}{\pi_{\text{ref}}(x_{t-1}^{(0)}|x_t,c)} - \beta\log\frac{\pi_\theta(x_{t-1}^{(1)}|x_t,c)}{\pi_{\text{ref}}(x_{t-1}^{(1)}|x_t,c)}
\end{aligned}
\tag{22}
$$

where $\mathbb{1}(\cdot)$ is the indicator function. The term $(-1)^{\mathbb{1}(r_t(c,x_{t-1}^{(0)})<r_t(c,x_{t-1}^{(0)}))}$ represents the step-level preference ranking procedure.

Furthermore, we compare TailorPO and TailorPO-G from the perspective of gradient, in order to understand their difference in effectiveness. In Eq. (11), we have shown that the gradient of the TarilorPO loss function can be written as follows.

$$
\nabla_\theta\mathcal{L}(\theta) = -\mathbb{E}\left[(f_t/\sigma_t^2)\cdot\nabla_\theta^T\mu_\theta(x_t)(x_{t-1}^w - x_{t-1}^l)\right]
$$

For TarlorPO-G, the term $x_{t-1}^w$ is modified by adding the gradient term $\nabla_{x_{t-1}^w}\log p(r_{\text{high}}|x_{t-1}^w)$. Therefore, we can derive its gradient term as follows.

$$
\begin{aligned}
\nabla_\theta\mathcal{L}_{TailorPO-G}(\theta) &= -\mathbb{E}\left[(f_t/\sigma_t^2)\cdot\nabla_\theta^T\mu_\theta(x_t)((x_{t-1}^w + \nabla_{x_{t-1}^w}\log p(r_{\text{high}}|x_{t-1}^w)) - x_{t-1}^l)\right] \\
&= -\mathbb{E}\left[(f_t/\sigma_t^2)\cdot\nabla_\theta^T\mu_\theta(x_t)(\underbrace{\nabla_{x_{t-1}^w}\log p(r_{\text{high}}|x_{t-1}^w)}_{\text{pushing towards high reward values}} + (x_{t-1}^w - x_{t-1}^l))\right]
\end{aligned}
\tag{23}
$$

The gradient term pushes the model towards the high-reward regions in the reward models. Therefore, TarlorPO-G further improves the effectiveness of TailorPO.

# C  FORMULATION OF THE DIFFUSION FRAMEWORK BASED ON EDM

In order to enhance the applicability of our study, in this section, we provide another formulation of diffusion models and our method using EDM (Karras et al., 2022).

Diffusion models contain a forward process and a reverse denoising process. Given an input $x_0$ sampled from the real distribution $p_{\text{data}}$, the forward process can be formulated as follows, which is a uniform formulation for DDPM, DDIM, and other methods.

$$x_t = s_t x_0 + s_t \sigma_t \epsilon,$$

where $x_t$ is the noisy sample at timestep $t$. $s_t$ represents a scale schedule coefficient and $\sigma_t$ represents the noisy schedule coefficient. At timestep $t$, we have $p(x_t|x_0) \sim \mathcal{N}(s_t x_0, s_t^2 \sigma_t^2 I)$. $s_t$ and $\sigma_t$ are usually selected to ensure the final output $x_T$ follows a certain Gaussian distribution.

The reverse process aims to recover the distribution of original inputs $x_0$ from a Gaussian noise $x_T$. According to [1], the reverse ODE process is given as follows.

$$dx = [\frac{\dot{s}_t}{s_t} x - s_t^2 \dot{\sigma}_t \sigma_t \nabla_x \log p(\frac{x}{s_t}; \sigma_t)] dt$$

where $\dot{s}_t$ and $\dot{\sigma}_t$ denote the time derivative. $\nabla_x \log p(\frac{x}{s_t}; \sigma_t)$ is the score function, which is usually approximated by a neural network, denoted by $s_\theta(\cdot)$. Replacing this term in the above equation, we can solve the reverse process. For a set of discrete timesteps, we can obtain a sequence $[x_T, x_{T-1}, \ldots, x_t, \ldots, x_0]$, and our study focuses on the optimization of $s_\theta(\cdot)$ at each timestep to generate $x_0$ with better image quality.

Subsequently, the predicted $\hat{x}_0$ at the step $t$ can be represented $\hat{x}_0(x_t) = \frac{1}{s_t}(x_t + s_t^2 \sigma_t^2 s_\theta(x_t))$. Then, the step-wise reward value of $x_t$ can be estimated based on $\hat{x}_0(x_t)$. Similarly, the conditional score function used in our gradient guidance can be rewritten as $\nabla_x \log p(\frac{x}{s_t}|r_{\text{high}}; \sigma_t) = \nabla_x \log p(\frac{x}{s_t}; \sigma_t) + \nabla_x \log p(r_{\text{high}}|\frac{x}{s_t}; \sigma_t)$. The first term is estimated by the neural network $s_\theta(\cdot)$, and the second term can be approximated following Eq. (13).

## D EXPERIMENTAL SETTINGS AND ETHICS STATEMENT FOR THE USER STUDY

To verify that our framework generates more human-preferred images, we conducted a user study by requesting ten human users to label their preference for generated images from the perspective of visual appeal and general preference.

**Ethics statement.** We collect feedback from ten annotators. All annotators acknowledge and agree that their efforts will be used to evaluate the performance of different methods in this paper.

**Task description.** Given each prompt in the set of 45 animal prompts, we sampled five images from the fine-tuned model and obtained a total of 225 images per model. For comparison, for each pair of fine-tuned model, we organized their generated images into 225 pairs. Each human annotator was given several triplets of $(c, x_1^{(a)}, x_0^{(b)})$, where $c$ is the text prompt and $x_1^{(a)}$ and $x_0^{(b)}$ represent the paired image generated by the model finetuned by method $a$ and method $b$, respectively. In order to avoid user bias, *we hid the source of $x_1^{(a)}$ and $x_0^{(b)}$ and randomly placed their order to annotators*. Then, the annotator was asked to compare the two images from the perspective of alignment, aesthetics, and visual pleasantness. If both images in a pair looked very similar or were both unappealing, then they should label "draw" for them. Otherwise, they should label each image with a "win" or "lose" tag. In this way, for each pair of comparing methods, we had 225 triplets of $(c, x_1^{(a)}, x_0^{(b)})$ and each annotator labeled 225 "win/lose" or "draw" tags.

Then, we computed the ratio of pairs where TailorPO and TailorPO-G received "win," "draw," and "lose" labels, respectively. Figure 7 reports the win-lose percentage results of our method versus other baseline methods, our method exhibits a clear advantage in aligning with human preference.

## E MORE EXPERIMENTAL RESUTLS

### E.1 VERIFICATION OF THE ESTIMATION FOR STEP-WISE REWARDS

In this section, we conducted experiments to verify the reliability of the estimation in Eq. (12) for step-wise rewards. We compared the estimated value $r(c, \hat{x}_0(x_t))$ with $r_t(c, x_t) \triangleq \mathbb{E}[r(c, x_0)|c, x_t]$ at different training checkpoints. For the fine-tuned model $\epsilon_{\theta'}$, we sampled images with 20 DDIM

steps and randomly sampled 100 pairs of $(c, x_t)$ at each timestep $t \in \{12, 8, 4, 1\}$. Give each pair of $(c, x_t)$, we sampled 100 images $x_0$ based on $x_t$ and then computed $r_t(c, x_t) = \mathbb{E}[r(c, x_0)|c, x_t]$ as the ground truth of the step-wise reward. Then, we computed the estimated value $r(c, \hat{x}_0(x_t))$ based on the fine-tuned parameters $\theta'$. Table **??** and Table **??** report the average relative error $\mathbb{E}[|\frac{r_t(c, x_t) - r(x, \hat{x}_0(x_t))}{r_t(c, x_t)}|]$ at different timesteps $t$ in different models (we used the aesthetic scorer and JPEG compressibility as the reward model, respectively).

Table 5: Average relative error of the estimated aesthetic score.

| timestep $t$ | 12 | 8 | 4 | 1 |
|---|---|---|---|---|
| pre-trained model $\epsilon_\theta$ | 0.0545±0.0427 | 0.0378±0.0287 | 0.0132±0.0089 | 0.0047±0.0051 |
| $\epsilon_{\theta'}$ after training on 10k samples | 0.0353±0.0345 | 0.0176±0.0160 | 0.0106±0.0080 | 0.0033±0.0029 |
| $\epsilon_{\theta'}$ after training on 40k samples | 0.1330±0.0320 | 0.0283±0.0231 | 0.0132±0.0084 | 0.0070±0.0047 |

Table 6: Average relative error of the estimated JPEG compressibility.

| timestep $t$ | 12 | 8 | 4 | 1 |
|---|---|---|---|---|
| pre-trained model $\epsilon_\theta$ | 0.2263±0.0524 | 0.1259±0.0333 | 0.0390±0.0101 | 0.0070±0.0039 |
| $\epsilon_{\theta'}$ after training on 10k samples | 0.2492±0.0390 | 0.1440±0.0279 | 0.0425±0.0071 | 0.0074±0.0016 |
| $\epsilon_{\theta'}$ after training on 40k samples | 0.1566±0.0925 | 0.0341±0.0221 | 0.0113±0.0077 | 0.0066±0.0016 |

These results demonstrate that after fine-tuning, the model $\epsilon_{\theta'}$ achieved a small error as the pre-trained model $\epsilon_\theta$ does. Moreover, our DPO-based loss function does not require an accurate reward value, but only needs the preference order of samples. Even if there is a small estimation error for the step-wise reward, it does not affect the preference order between paired samples, thus having little effect on training. Therefore, the estimation for step-wise rewards is reliable.

### E.2 EXPERIMENTS ON COMPLEX PROMPTS

We fine-tuned Stable Diffusion v1.5 on various reward models using 4k prompts in the training set of the Pick-a-Pic validation set (Kirstain et al., 2023), selected by Liang et al. (2024). We followed the same setting with Section 4 of the main text for TailorPO and TailPO-G. Then, we evaluated the fine-tuned model on 500 prompts from the Pick-a-Pic validation set. Table 7 compares our method with Diffusion-DPO (Wallace et al., 2024) and SPO (Liang et al., 2024)[1]. For these complex prompts, our methods also achieved the highest reward values.

Table 7: Reward values of images generated by diffusion models fine-tuned using different methods. The prompts are from the Pick-a-Pic dataset.

| | Diffusion-DPO | SPO | TailorPO | TailorPO-G |
|---|---|---|---|---|
| Aesthetic scorer | 5.505 | 5.887 | 6.050 | 6.242 |
| ImageReward | 0.1115 | 0.1712 | 0.3820 | 0.3791 |

### E.3 EXPERIMENTS ON OTHER BASE MODELS

We also fine-tuned Stable Diffusion v2.1[2] (SD-v2.1-base) to demonstrate the effectiveness of our method. Taking the aesthetic scorer as the reward model, we fine-tuned SD-v2.1-base using prompts of animals, and then evaluated the model with the same prompts. After fine-tuning with TailorPO, the aesthetic score of images generated by SD-v2.1-base was improved from 5.95 to 6.21. In comparison, DDPO only reached the value of 6.02.

### E.4 RESULTS OF TRAINING ON MORE PAIRED SAMPLES

We compared the performance of different methods when training on more paired samples. We took the aesthetic scorer as the reward model to fine-tune SD v1.5 and SD v2.1-base using DDPO and TailorPO on 40k paired samples, and we took the JPEG compressibility as the reward to fine-tune

---

[1]Results of Diffusion-DPO and SPO on prompts in Pick-a-Pic dataset are from (Liang et al., 2024).

[2]https://huggingface.co/stabilityai/stable-diffusion-2-1-base

SD v1.5 using DDPO and D3PO on 30k paired samples. Figure 9 and Figure 10 show the change in the reward value during the training process. We observed three phenomena from these results.

- TailorPO increased reward values the most effectively within the least number of paired samples. This means that we can use fewer samples than other methods to achieve a good performance.

- TailorPO achieved and even surpassed the best performance of DDPO and D3PO on JPEG compressibility.

- Although DDPO reached the highest aesthetic score at 40k samples on SD v1.5, we observed a severe reward hacking problem with generated images. We provide some examples in Figure 9. All these images are unnatural with the same color, same style, and similar background (yellow leaves). Therefore, instead of fine-tuning the diffusion model with too many samples to achieve an extremely high reward score, we would suggest controlling the number of samples to strike a balance between good image quality and a high reward score.

- D3PO was less effective than both DDPO and TailorPO, and this conclusion was consistent with Figure 3 of D3PO's original paper. This phenomenon also supports our discovery of its inherent issues about preference order and gradient direction.

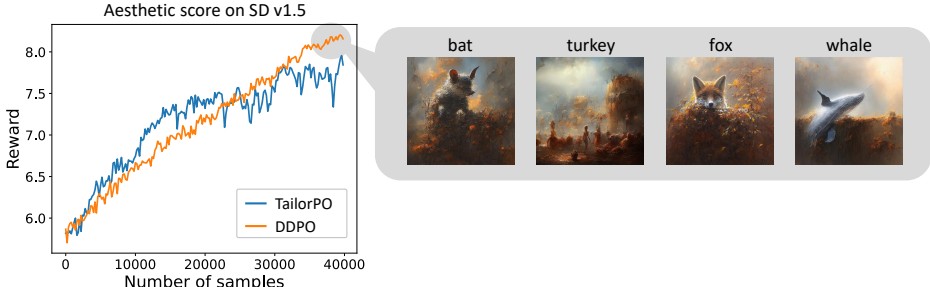

Figure 9: The change curve of the aesthetic score during the fine-tuning process on SD v1.5.

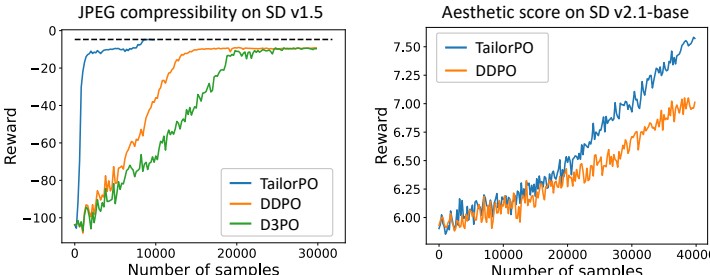

Figure 10: The change curve of JPEG compressibility and aesthetic score during the fine-tuning process of SD v1.5 and SD v2.1-base.

### E.5 GENERATIONS GIVEN DIFFERENT REWARD MODELS AND PROMPTS.

In this section, we provide some examples of generated images given different reward models and prompts from the main text. For different models, Figure 11 shows images generated by SD-v1.5 fine-tuned on the PickScore reward model. For different prompts, we designed and selected[3] several real-world prompts, which were not presented in the training set of prompts. Figure 12 shows that the model generated natural and beautiful images accordingly.

---

[3]We selected several prompts from https://openai.com/index/dall-e-3/.

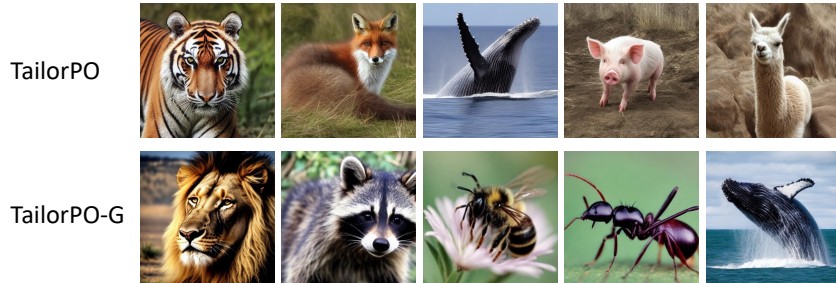

Figure 11: Images generated by the model fine-tuned on the PickScore reward model.

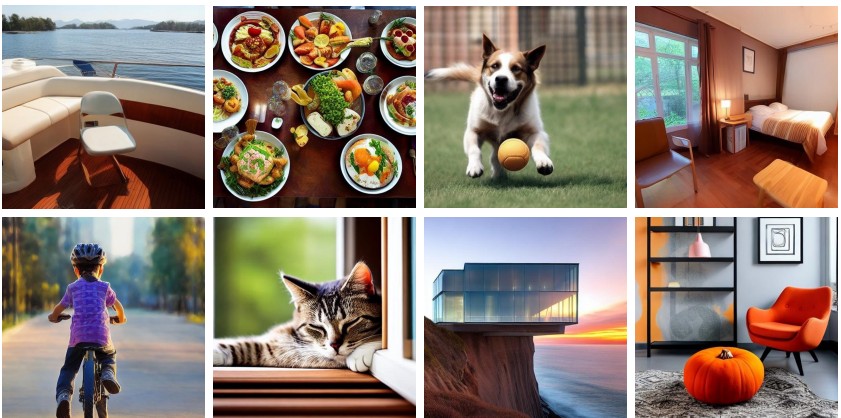

Figure 12: Images generated given real-world prompts: (1) A chair in the corner on a boat. (2) A table of delicious food. (3) A dog playing a ball. (4) A warm and comfortable room with a table, a chair, and a bed. (5) A kid riding a bike. (6) A cat sleeping next to the window. (7) A modern architectural building with large glass windows, situated on a cliff overlooking a serene ocean at sunset. (8) Illustration of a chic chair with a design reminiscent of a pumpkin's form, with deep orange cushioning, in a stylish loft setting.

## F  ABLATION STUDIES

In this section, we performed ablation studies to verify the effect of hyper-parameters on performance, including the number of steps used for optimization and the strength of gradient guidance. Furthermore, we investigated the impact of each component in our framework.

**Effect of steps used for training.** We first investigate the effect of the number of steps $T_{\text{fine-tune}}$ used for fine-tuning in TailorPO. In Section 4, We generated images with $T = 20$ sampling timesteps and uniformly sampled only $T_{\text{fine-tune}} = 5$ steps for training to boost the training efficiency. Here, we compared the results of setting $T_{\text{fine-tune}} = 3, 5, 10$ in Table 8, and it shows that while the fine-tuning performance is relatively stable to the setting of $T_{\text{fine-tune}}$, fine-tuning on five steps achieved a better trade-off between performance and efficiency.

**Effect of the strength of gradient guidance.** We also verify the effect of gradient guidance in TailorPO-G by applying gradient guidance with different strengths at intermediate steps. Specifically, we used different settings of $\eta_t$ in Eq. (14) for fine-tuning. The result in Table 9 shows that the varying strength $\eta_t$ for different steps $t$ better enhanced the fine-tuning performance.

**Effects of each component in our methods.** There are three key components in our methods: (1) step-level preference ranking, (2) the same input condition at each step, and (3) gradient guidance of reward models. Therefore, we fine-tuned SD-v1.5 based on the aesthetic scorer using (1), (1)+(2), (1)+(2)+(3). Here we set the same random seed for a fair comparison, so the results of (1)+(2) and (1)+(2)+(3) were slightly different from Table 2 (where we averaged results of three runs with different random seeds). Table 10 shows that all these components improved the aligning effectiveness.

Table 8: Effect of the number of steps used in TailorPO. For each setting of $T_{\text{fine-tune}}$, we uniformly sampled $T_{\text{fine-tune}}$ steps for fine-tuning.

| $T_{\text{fine-tune}}$ | Aesthetic Scorer | HPSv2 | compressibility |
|---|---|---|---|
| 10 | 6.61 | 28.14 | -20.62 |
| 5 | **6.74** | **28.43** | **-4.76** |
| 3 | 6.40 | 28.15 | -9.97 |

Table 9: Effect of strength $\eta_t$ of gradient guidance in TailorPO-G. [0.1,0.2] represents we set $\eta_t$ ranging from 0.1 to 0.2 for different $t$.

| $\eta_t$ | Aesthetic Scorer | ImageReward | HPSv2 |
|---|---|---|---|
| 0.1 | 5.82 | 1.22 | 28.10 |
| 0.2 | 6.97 | **1.35** | 28.18 |
| 0.5 | 7.07 | 0.71 | 27.48 |
| [0.1, 0.2] | **7.11** | 1.25 | **28.43** |

Table 10: Effect of each component in our framework.

| | Aesthetic scores | ImageReward |
|---|---|---|
| SD-v1.5 | 5.79 | 0.65 |
| (1) step-level preference ranking | 6.40 | 0.98 |
| (1) step-level preference ranking + (2) same input condition at each step | 6.69 | 1.16 |
| (1) step-level preference ranking + (2) same input condition at each step + (3) gradient guidance | 6.78 | 1.25 |

# G LIMITATIONS AND DISCUSSIONS

In this section, we discuss the potential limitations of our method. Like other methods based on an explicit pre-trained reward model, including DDPO, D3PO, and SPO, TailorPO has the potential of being prone to reward hacking (Skalse et al., 2022), if we fine-tune the model on very simple prompts for too many iterations. It means that the generative model is overoptimized to improve the score of the reward model but fails to maintain the original output distribution of natural images. We provide some examples in Figure 13 to demonstrate this phenomenon.

Figure 13: Fine-tuning the diffusion model based on pre-trained reward models may introduce some bias into the generated images. For example, when taking JPEG compressibility as the reward model, DDPO, D3PO, and our methods all generate images with a blank background.

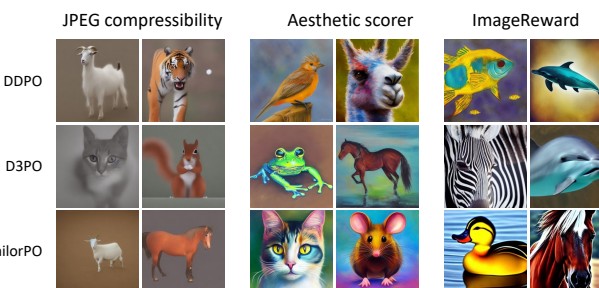

The problem of reward hacking is related to the quality of reward models. Given the fact that these pre-trained reward models are usually trained on a finite training set, they cannot *perfectly* fit the human preference for natural and visually pleasing images. Therefore, the optimization of generative models towards these reward models may lead to an unnatural distribution of images.

In order to alleviate the reward hacking problem, TailorPO can be further improved from the following perspectives.

- Using a better reward model that well captures the distribution of natural and visually pleasing images. A better reward model can avoid guiding model optimization towards unnatural images.

- Utilizing the ensemble of multiple reward models to alleviate the bias of a single reward model. While each single reward model has its own preference bias, considering multiple reward models altogether may be able to alleviate the risk of falling into a single model. To this end, Coste et al. (2024) have shown that the reward model ensembles can effectively address reward hacking in RLHF-based fine-tuning of language models. Therefore, we are hopeful that the reward model ensembles are also effective for diffusion models.

- Searching for a better setting of the hyperparameter $\beta$ in the loss function to strike a balance between natural images and high reward scores. In DPO-style methods, the coefficient $\beta$ controls the deviation from the original generative distribution (the KL regularization). In this way, we can search for a better value of $\beta$ to avoid the model being fine-tuned far away from the original base model. For example, Wu et al. (2024) have provided a method to dynamically adjust the value of $\beta$.

