# OpenReview forum: "A Tailored Framework for Aligning Diffusion Models with Human Preference"
_ICLR.cc/2025/Conference — Submitted to ICLR 2025_

### Official Review · Reviewer_ZmZ1 · 2024-11-01

**Soundness:** 2
**Presentation:** 3
**Contribution:** 3
**Rating:** 6
**Confidence:** 3

**Summary:**

This paper identifies a key weakness in the formation of existing framework that aligns diffusion model with human preference.  The authors noted that if the winning and losing samples are in a linear subspace, it is possible for the gradient updates to take the wrong direction. To address this issue, the authors propose a novel tailored framework which ensures the correct update.  Experiments show that the proposed modification improves the performance of alignment algorithms on a variety of reward functions,

**Strengths:**

1. The presentation is clear and easy to follow. The problem is well motivated, and concrete derivations are provided.
2. The authors conducted extensive experiments on a wide range of prompts and reward target to highlight the effectiveness of the method
3. The authors provided user study results, which are helpful to contextualize the implications of qualitative results.

**Weaknesses:**

1. The details of user study are not disclosed and results are not carefully analyzed. While the author provide some overviews in Appendix B, they did not disclose how many user responses are collected from the 225 images generated. This is important because it determines the standard error and confidence interval of the user study, which is crucial to judge the significance of the results. The author also did not disclose the instructions provided to the user, and if proper mitigations are applied to reduce user biases (e.g. randomize image order of different models).

2. While the author showed that the proposed method works well in comparison with respect, to other online methods, but fails to compare against state-of-the art offline methods such as Diffusion-DPO. Online methods are costly in that they require ad-hoc sampling from the diffusion model. Such complexity should be justified.  In particular, Diffusion-DPO also uses Pick-a-Pic dataset to align for human preference.

**Questions:**

See weakness

Addition question that did not affect the decision:

1. What are the performance on in-distribution prompts? In particular, Table 3 shows that TailorPO-G loses to TailorPO on HPSv2, which is surprising because one would expect with direct gradient guidance from the reward model,  TailorPO-G  should perform better. Is this because TailorPO-G overfits to the training prompts?

---

> ### Author Response · Authors · 2024-11-20
> **Response to Reviewer ZmZ1**
>
> Thank you for your positive feedback and valuable comments. We will try our best to answer all your concerns. Please let us know if you still have further concerns, so that we can further update the response ASAP.
>
> **Q1: About details of user study. "The details of user study are not disclosed and results are not carefully analyzed."**
>
> **A1:** Thank you.  We would like to provide more details about the user study, and we have followed your suggestion to add these details in Appendix D of the revised manuscript.
>
> - Participant: We collect feedback from five annotators in the original manuscript, and we include five more annotators during the rebuttal phase. All annotators acknowledge that their efforts will be used to evaluate the performance of different methods in this paper.
> - Task instruction: The human annotators are given several triplets of ($c, x^{(a)}_1, x^{(b)}_0$), where $c$ is the text prompt and $x^{(a)}_1$ and $x^{(b)}_0$ represent the image generated by the model finetuned by method $a$ and method $b$, respectively. Then, the annotator is asked to compare the two images from the perspective of alignment, aesthetics, and visual pleasantness. If both images in a pair look very similar or are both unappealing, then the annotator should label “draw” for them. Otherwise, they label the "win" and "lose" tag for each image. In this way, for each pair of comparing methods, we have 225 triplets of ($c, x^{(a)}_1, x^{(b)}_0$) and each annotator label 225 "win/lose" or "draw" tags.
> - Mitigation: In order to avoid user bias, we hide the source of $x^{(a)}_1$ and $x^{(b)}_0$ and randomly place their order to annotators.
>
> ---
>
> **Q2: About comparison with Diffusion-DPO. "fails to compare against state-of-the art offline methods such as Diffusion-DPO."**
>
> **A2:**  Thank you. We have followed your suggestion to *conduct a new experiment* to compare TailorPO with Diffusion-DPO (Wallace et al., 2024). Diffusion-DPO finetuned SD-v1.5 on the Pick-a-Pic dataset in an offline manner. Therefore, we also finetune SD-v1.5 using TailorPO on prompts in the Pick-a-Pic training set and evaluate the performance using prompts in the Pick-a-Pic validation set. We use aesthetic scorer and ImageReward as the reward model, respectively.
>
> |               | Aesthetic score | ImageReward |
> | ------------- | --------------- | ----------- |
> | Diffusion-DPO | 5.505           | 0.1115      |
> | TailorPO      | 6.050           | 0.3820      |
> | TailorPO-G    | 6.242           | 0.3791      |
>
> The results of Diffusion-DPO in the above table are from (Liang et al., 2024). The above table shows that our methods achieve higher reward values than Diffusion-DPO in both aesthetic score and ImageReward score.
>
> ---
>
> **Q3: About performance on in-distribution prompts. "What are the performance on in-distribution prompts? Table 3 shows that TailorPO-G loses to TailorPO on HPSv2 ... Is this because TailorPO-G overfits to the training prompts?"**
>
> **A3:**  I would like to confirm whether the "in-distribution prompts" refer to prompts used in training. If so, then Table 2 reports the performance on in-distribution prompts. Furthermore, we have *conducted new experiments* to finetune and evaluate SD-v1.5 on complex prompts in the Pick-a-Pic dataset. In this case, prompts used for evaluations are also in-distribution prompts.
>
> |            | Aesthetic score | ImageReward |
> | ---------- | --------------- | ----------- |
> | SD-v1.5    | 5.69            | -0.04       |
> | TailorPO   | 6.05            | 0.38        |
> | TailorPO-G | 6.24            | 0.38        |
>
> On the other hand, In Table 3, the model is finetuned on 45 simple prompts of animals and tested on 500 complex prompts in the Pick-a-Pic dataset. In this case, the testing prompts contain many descriptions of objects, scenes, light, and style, and these descriptions are unseen in finetuning. Therefore,  TailorPO-G may only strengthen the ability of the model to generate high-quality animal-related objects but cannot cover all these complex scenes. Nevertheless, both TailorPO and TailorPO-G have outperformed previous methods in most cases.

---

> > ### Comment · Reviewer_ZmZ1 · 2024-11-27
> >
> > The authors have successfully addressed my concerns. I especially appreciate the new results with Diffusion-DPO. The proposed method clearly outperforms offline methods such as Diffusion-DPO.

---

> > > ### Author Response · Authors · 2024-11-28
> > >
> > > Thank you very much for your reply! We sincerely appreciate your valuable comments that have helped us improve our paper.

---

### Official Review · Reviewer_F2pk · 2024-11-01

**Soundness:** 2
**Presentation:** 3
**Contribution:** 2
**Rating:** 6
**Confidence:** 3

**Summary:**

This paper proposed TailorPO, a DPO framework tailored for diffusion model based on previous D3PO method. The framework includes three key improvement: 1) Turn the preference ranking on step-level instead of final image level; 2) The preference is only considered on the same condition; 3) Use gradient guidance to increase the difference on the reward between each pair.

**Strengths:**

- The paper is well-written and effectively organized. The authors provide a thorough analysis of the limitations in existing DPO methods, including challenges with inaccurate preference ordering and gradient direction, and subsequently propose a method that addresses these issues effectively.
- The proposed method is straightforward to understand, with clear an accesible theoretical analysis, particularly in its comparison of formulations with prior methods such as D3PO.
- The experiments examine the generalization capability of the proposed method across different prompts and reward models, a crucial aspect for fine-tuning approaches.

**Weaknesses:**

- The paper lacks a crucial ablation study on the contribution of each component in TailorPO to its effectiveness. Specifically, it would be valuable to evaluate the individual impact of (1) step-level preference, (2) preference indication only under the same condition, and (3) gradient guidance. This analysis is essential to substantiate the contribution of the overall framework beyond that of its individual components.
- The paper lacks verification of generalization on fine-tuning methods, such as LoRA and full fine-tuning. Additionally, the experiments appear to be conducted solely on SD 1.5; expanding evaluation to include a broader range of base models would strengthen the results.
- Figure 5 indicates an evident over-saturation issue for both TailorPO and TailorPO-G, which does not align with the caption’s description of producing "more visually pleasing images." Conducting a user study could better substantiate this claim.
- The framework may not be data-efficient, as it seems unable to leverage existing dataset including preference pairs of candidates across different conditions.

**Questions:**

- It may be beneficial to reflect the preference selection procedure in Eq.10, as this equation is regarded as the formulation of the TailorDPO framework.
- Could you provide insights into the differences between TailorOP and TailorOPG?
- I am concerned about the method's performance in real-world applications. Why do TailorPO and TailorPO-G lead to over-saturation issues in Figures 5 and 6?
- The baseline for comparison seems relatively weak (with timesteps set to 20 for DDIM, which is uncommon in practice). If available, could you share results using the typical choice of 50 timesteps for DDIM or 25 timesteps for other advanced schedulers?

I'm willing to increase my rating if the concerns can be addressed.

---

> ### Author Response · Authors · 2024-11-20
> **Response to Reviewer F2pk (part 1)**
>
> Thank you for your valuable comments. We will try our best to answer all your concerns. Please let us know if you still have further concerns, so that we can further update the response ASAP.
>
> **Q1: About ablation study. "ablation study on the contribution of each component in TailorPO to its effectiveness."**
>
> **A1:** Thank you. We have followed your suggestion to *conduct a new experiment* on the contribution of each component in TailorPO and TailorPO-G. There are three key components: (1) step-level preference ranking, (2) the same input condition at each step, and (3) gradient guidance of reward models. Therefore, we fine-tune SD-v1.5 based on the aesthetic scorer using  (1), (1)+(2), (1)+(2)+(3). Here we set the same random seed for a fair comparison, so the results of (1)+(2) and (1)+(2)+(3) are slightly different from Table 2 (where we averaged the results of three runs under different random seeds). The following table shows that all these components improve the aligning effectiveness. We have added this experiment to Appendix F of the revised manuscript.
>
> |                                                              | Aesthetic scorer | ImageReward |
> | ------------------------------------------------------------ | ---------------- | ----------- |
> | SD-v1.5                                                      | 5.79             | 0.65        |
> | (1) step-level preference ranking                            | 6.40             | 0.98        |
> | (1) step-level preference ranking + (2) same input condition at each step | 6.69             | 1.16        |
> | (1) step-level preference ranking + (2) same input condition at each step + (3) gradient guidance | 6.78             | 1.25        |
>
> ---
>
> **Q2: About generalization on different fine-tuning methods and base models. "verification of generalization on fine-tuning methods, such as LoRA and full fine-tuning ... expanding evaluation to include a broader range of base models would strengthen the results."**
>
> **A2:** (1) Generalization on different fine-tuning methods. In this study, we use LoRA because almost all fine-tuning methods for aligning diffusion models used LoRA, including DPOK, DDPO, D3PO, SPO, and DenseReward. For a fair comparison, we also fine-tuned the model using LoRA. On the other hand, due to the limited resources, we are not able to fully fine-tune a diffusion model in an acceptable period of time. Nevertheless, we are positive about the generalization ability of our method on different fine-tuning methods, given that it has demonstrated effectiveness on LoRA.
>
> (2) Generalization on different base models. We have followed your suggestion to *conduct a new experiment* on Stable Diffusion-v2.1-base (SD-v2.1-base, https://huggingface.co/stabilityai/stable-diffusion-2-1-base). We fine-tune SD-v2.1 on the set of animal-related prompts by taking the aesthetic scorer as the reward model, and then evaluate the model using the same prompts. After fine-tuning with TailorPO, we improve the aesthetic score of generated images from 5.95 to 6.21. In comparison, DDPO only reaches 6.02.
>
> ---
>
> **Q3: About user study. "Figure 5 indicates an evident over-saturation issue ... Conducting a user study could better substantiate this claim."**
>
> **A3:** Thank you. We have conducted a user study on these generation results as you requested, as stated in Lines 469-473 of the original manuscript (Lines 469-475 of the revised manuscript). Results in Figure 7 show that TailorPO and TailorPO-G receive higher preference than previous methods. Moreover, in the revised manuscript, we extend the user study and collect feedback from a total of ten human annotators. The result in Figure 7 shows that our method indeed generated human-preferred images.
>
> On the other hand, the over-saturation issue in Figure 5 is caused by the overoptimization of the model towards the preference bias of the reward model. We provide a detailed discussion about this problem in the answer to your Q7. When we used other reward models, Figure 10 in Appendix E.4 of the revised manuscript demonstrates that the over-saturation issue does not appear.

---

> ### Author Response · Authors · 2024-11-20
> **Response to Reviewer F2pk (part 2)**
>
> **Q4: "The framework may not be data-efficient, as it seems unable to leverage existing dataset including preference pairs of candidates across different conditions."**
>
> **A4:** This is a good question. Although our method cannot directly leverage existing datasets in an offline manner, the online learning of our method has additional advantages, including good performance and good generalization ability.
>
> First, many studies [1-5] have shown that online strategies of learning algorithms significantly outperform their offline counterparts, while offline strategies face the challenges of OOD samples and gradient issues [4]. The following table also shows that our methods outperform the state-of-the-art offline method, Diffusion-DPO  (Wallace et al., 2024). The results of Diffusion-DPO in the following table are from (Liang et al., 2024).
>
> |               | Aesthetic score | ImageReward |
> | ------------- | --------------- | ----------- |
> | Diffusion-DPO | 5.505           | 0.1115      |
> | TailorPO      | 6.050           | 0.3820      |
> | TailorPO-G    | 6.242           | 0.3791      |
>
> Second, our method has good generalization ability. The online TailorPO is applicable to open-vocabulary scenarios, not limited by prompts in the dataset. In addition, Table 3 and Figure 8 have shown that our methods exhibit good generalization ability over different prompts. Besides, TailorPO-G incorporates the gradient guidance of reward models in the framework, which supports the injection of different conditions in training.
>
> Finally, if we want to leverage an existing dataset for TailorPO, the most straightforward approach is to first train a reward model on the given dataset and then align the diffusion model towards the reward model. In fact, the training of the reward model is also effective. For example, [6] has found that a simple linear model on the top of CLIP ViT/14 is sufficient to produce satisfying results.
>
> [1] Liu et al., Statistical Rejection Sampling Improves Preference Optimization. ICLR 2024.
>
> [2] Xu et al., Is DPO Superior to PPO for LLM Alignment? A Comprehensive Study. ICML 2024.
>
> [3] Xiong et al., Iterative Preference Learning from Human Feedback: Bridging Theory and Practice for RLHF under KL-constraint. arXiv: 2312.11456.
>
> [4] Feng et al., Towards Analyzing and Understanding the Limitations of DPO: A Theoretical Perspective. arXiv: 2404.04626.
>
> [5] Dong et al., RLHF Workflow: From Reward Modeling to Online RLHF. TMLR 2024.
>
> [6] Schuhmann et al., LAION-5B: an open large-scale dataset for training next generation image-text models. NeurIPS 2022.
>
> ---
>
> **Q5: "reflect the preference selection procedure in Eq.10"**
>
> **A5:** Thank you for the kind suggestion. We did not explicitly include the preference selection procedure in Eq. (10) because Eq. (10) followed the classic formulation of DPO (Eq. (3)) and we thought it was easy to be understood by readers.  On the other hand, the suggestion of "reflecting the preference selection procedure in Eq.10" is also insightful, and we rewrite the loss function in Eq. (10) as follows.
>
> $\mathcal{L}(\theta) = -\mathbb{E}\_{(c,x\_{t},x^{(0)}\_{t-1}, x^{(1)}\_{t-1})} \left[\log \sigma \left((-1)^{\mathbb{1}(r\_t(c,x^{(0)}\_{t-1})<r\_t(c,x^{(1)}\_{t-1}))} \cdot \left[\beta\log \frac{\pi\_\theta(x^{(0)}\_{t-1}|x\_{t}, c)}{\pi_\text{ref}(x^{(0)}\_{t-1}|x\_{t}, c)} - \beta\log \frac{\pi\_\theta(x^{(1)}\_{t-1}|x\_{t}, c)}{\pi\_\text{ref}(x^{(1)}\_{t-1}|x\_{t}, c)} \right]\right)\right]$
>
> where $\mathbb{1}(\cdot)$ is the indicator function. The term $(-1)^{\mathbb{1}(r\_t(c,x^{(0)}\_{t-1})<r\_t(c,x^{(0)}\_{t-1})}$ represents the step-level preference ranking procedure. We have added this form of the loss function in Appendix B of the revised manuscript for clarification.

---

> ### Author Response · Authors · 2024-11-20
> **Response to Reviewer F2pk (part 3)**
>
> **Q6: "provide insights into the differences between TailorPO and TailorPO-G"**
>
> **A6:** The key difference between TailorPO and TailorPO-G is that the gradient guidance better aligns the optimization direction with the reward model. We will elaborate on this by analyzing the gradient in TailorPO and TailorPO-G. In Eq.(11) of the original manuscript, we have shown that the gradient of the TarilorPO loss function can be written as follows.
> $$
> \nabla_\theta \mathcal{L}(\theta) = -\mathbb{E}\left[(f_t/{\sigma^2_{t}})\cdot\nabla^T_\theta\mu_\theta(x_{t})(x^w_{t-1} - x^l_{t-1})\right]
> $$
> For TarlorPO-G, the term $x^w_{t-1}$ is modified by adding the gradient term $\nabla_{x^w_{t-1}}\log p(r_{\text{high}}|x^w_{t-1})$. Therefore, we can derive its gradient term as follows.
> $$
> \begin{array}{ll}
> \nabla\_\theta \mathcal{L}\_{TailorPO-G}(\theta) & = -\mathbb{E}\left[(f\_t/{\sigma^2\_{t}})\cdot\nabla^T\_\theta\mu\_\theta(x\_{t})((x^w\_{t-1} +  \nabla\_{x^w\_{t-1}}\log p(r\_\text{high} | x^w\_{t-1}))- x^l\_{t-1})\right] \\
> &= -\mathbb{E}\left[(f\_t/{\sigma^2\_{t}})\cdot\nabla^T\_\theta\mu\_\theta(x\_{t})(\underbrace{\nabla\_{x^w\_{t-1}}\log p(r\_\text{high} | x^w\_{t-1})}\_{\text{pushing towards high reward values}} + (x^w\_{t-1} - x^l\_{t-1})\right]
> \end{array}
> $$
> The gradient term pushes the model towards the high-reward regions in the reward models. Therefore, TarlorPO-G further improves the effectiveness of TailorPO. Thank you for your insightful comment and we have added this discussion in Appendix B of the revised manuscript.
>
> ---
>
> **Q7: "I am concerned about the method's performance in real-world applications. Why do TailorPO and TailorPO-G lead to over-saturation issues in Figures 5 and 6?"**
>
> **A7:** To address your concern, we first evaluate the method's performance in real-world prompts. We design and sample several prompts from [1], and generate images using the model fine-tuned by our methods. Please refer to Figure 11 of the revised manuscript for visual demonstrations. These results show that on real-world prompts, the generated images are natural and have good quality,  not exhibiting the over-saturation issue.
>
> Second, we would like to discuss the over-saturation issue in Figure 5/6. This issue is caused by that the generative model is overoptimized to improve the score of the reward model on a given training set of prompts, but a high reward score may cause the generation distribution shift.  We provide some examples in Figure 12 of the revised manuscript to demonstrate this phenomenon. For example,  when taking JPEG compressibility as the reward model, DDPO, D3PO, and our methods all generate images with a blank background. In Figure 5 and Figure 6, the reward models are ImageReward and aesthetic scorer, which are trained on human preference rankings and potentially prefer images with bright colors (as shown in Figure 1 of (Xu et al., 2023)). In contrast, if we use other reward models (Figure 10), or use real-world prompts not in the training set (Figure 11), the over-saturation issue does not appear.
>
> Nevertheless, the over-saturation issue in Figure 5 and Figure 6 is in an acceptable range. These figures are colorful and contain more details, but have no distortion.  The user study in Figure 7 also validates that our method generates more preferred images.
>
> [1] https://openai.com/index/dall-e-3/

---

> ### Author Response · Authors · 2024-11-20
> **Response to Reviewer F2pk (part 4)**
>
> **Q8: About the baseline for comparison. "The baseline for comparison seems relatively weak ... share results using the typical choice of 50 timesteps for DDIM or 25 timesteps for other advanced schedulers"**
>
> **A8:** Thank you for the suggestion. We have followed your suggestion to *conduct a new experiment* to compare TailorPO with more baselines. We measure the reward values of images generated by the base model using 50 and 100 DDIM timesteps and 50 timesteps for PDNM and DPM++ schedulers, respectively. We also fine-tune and evaluate the model using DDPO and D3PO with 50 timesteps for DDIM, respectively. The following table reports the results of the base model using different settings, as well as the results of our methods using 20 timesteps for DDIM. Our methods still outperform the strengthened baselines.
>
> |                                     | Aesthetic Scorer | ImageReward | HPSv2     | PickScore | Compressibility |
> | ----------------------------------- | --------------- | ----------- | --------- | --------- | --------------- |
> | SD-v1.5, 20 timesteps for DDIM      | 5.79            | 0.65        | 27.51     | 20.20     | -105.51         |
> | SD-v1.5, 50 timesteps for DDIM      | 5.81            | 0.80        | 27.69     | 20.24     | -108.67         |
> | SD-v1.5, 100 timesteps for DDIM     | 5.79            | 0.83        | 27.73     | 20.18     | -111.46         |
> | SD-v1.5, 50 timesteps for PNDM  | 5.64            | 0.67        | 27.51     | 20.12     | -123.68         |
> | SD-v1.5, 50 timesteps for DPM++ | 5.64            | 0.70        | 27.57     | 20.10     | -123.71         |
> | DDPO, 50 timesteps for DDIM         | 6.65            | --          | --        | --        | --              |
> | D3PO, 50 timesteps for DDIM         | 6.37            | --          | --        | --        | --              |
> | TailorPO                            | 6.66            | 1.20        | **28.37** | 20.34     | **-6.71**       |
> | TailorPO-G                          | **6.96**        | **1.26**    | 28.03     | **20.68** | -               |

---

> > ### Comment · Reviewer_F2pk · 2024-11-25
> >
> > Thank you for the detailed discussion and the extensive experimental results. Most of my concerns have been addressed. I also reviewed the feedback provided by other reviewers. Based on the ablation study, it is evident that the contribution preference among the components is *step-awared preference* (0.61 $\uparrow$) > *same conditions* (0.29 $\uparrow$) > *gradient guidance to enlarge difference* (0.09 $\uparrow$).
> >
> > I understand that you chose a different reward function compared to SPO. However, this distinction should be emphasized and explicitly reflected in your main paper, particularly in Section 3.2 which serves as the main motivation for the whole method. Given that the theoretical analysis of the necessity of step-aware optimization is presented as one of your most significant contributions, it would be beneficial to include a separate discussion comparing your approach with SPO. Specifically, you should explain your improvements over SPO based on your analysis (e.g., why 'directly estimating the step-wise reward' is better than 'using an additional step-awared reward model'). This would provide a more comprehensive understanding of the novelty and significance of the proposed framework while acknowledging the contribution from previous work.
> >
> > I strongly recommend adding this discussion in introduction / section 3.2 / appendix to improve the overall clarity and impact of the paper and address the integrity concern from other reviewers.

---

> > > ### Author Response · Authors · 2024-11-25
> > >
> > > Thank you very much for your helpful suggestion. We would like to first clarify that our difference is not limited to the "reward function". First, we have distinct motivations based on our theoretical analysis. Second, we propose a different and novel method to evaluate the step-level reward. Third, we are directly inspired by the difference between Eq. (3) and Eq. (5) to set the same input $x_t$ at each step, and we prove this operation could align the optimization direction with preferred samples. Fourth, we introduce the gradient guidance of reward models into the aligning framework to further improve the effectiveness.
> > >
> > > Second, besides the discussions about our difference from SPO in Lines 126-130 and Lines 204-209 of the original manuscript, we have followed your suggestion to add more discussions to further explain our differences in the introduction, Section 3.2, 3.3, 4.1, as follows.
> > >
> > > - In Introduction: "Most close to our work, Liang et al. (2024) also noticed the inconsistency of the preference order between intermediate-step outputs and final images, and they proposed to train an additional step-wise reward model to address this issue.  In comparison, we are the first to explicitly derive the theoretical flaws of previous DPO implementations in diffusion models, and we propose distinct solutions to address these issues. Experiments also demonstrate that our framework outperforms SPO on various reward models."
> > >
> > > - In Section 3.2: "In this section, we have conducted a detailed analysis of the denoising process based on MDP, and the optimization gradient of diffusion models. In this way, we reveal the potential theoretical issues in previous methods beyond visual discoveries of (Liang et al., 2024). To address these issues, we propose distinct solutions in Section 3.3"
> > >
> > > - In Section 3.3: "Different from (Liang et al., 2024), we aim to address the gradient issue in Section 3.2 and it is straightforward to sample from the same $x_t$ based on our theoretical analysis." "To this end, Liang et al. (2024) proposed to train a step-wise reward model based on another uncertified assumption, *i.e.*, 'the preference order between pair of images can be kept when adding the same noise.' In comparison, we directly evaluate the preference quality of noisy samples $x_t$ without training a new model. "
> > >
> > > - In Section 4.1: "Notably, our methods outperform SPO as we directly estimate the step-level reward without training another reward model based on an uncertified assumption, and we incorporate the gradient guidance to further improve the effectiveness."
> > >
> > > In addition, we would like to clarify that the contribution of gradient guidance is actually larger than 0.09. According to results in Table 2 and Figure 4, the gradient guidance improves the aesthetic score of images from 6.66 to 6.96, and it improves the PickScore from 20.34 to 20.68, which is averaged on three runs. This improvement is related to the reward model and is affected by randomness in the experiment, so the result in our rebuttal (Table 8) demonstrates a smaller effect.
> > >
> > > We hope that our response could adequately address all your concerns, and we sincerely hope you can reconsider the rating accordingly.

---

> > > > ### Comment · Reviewer_F2pk · 2024-11-25
> > > >
> > > > Thank you for your comprehensive revision. I have adjusted my score accordingly. However, as noted in my previous review and other reviewers, a dedicated section discussing the differences would further enhance the clarity of your work. For this reason, I will maintain my confidence score at 3.

---

> > > > > ### Author Response · Authors · 2024-11-25
> > > > >
> > > > > Thank you very much! We would consider your suggestion to better clarify the differences in the main text and the appendix.

---

### Official Review · Reviewer_zqb2 · 2024-11-03

**Soundness:** 2
**Presentation:** 2
**Contribution:** 1
**Rating:** 1
**Confidence:** 5

**Summary:**

DPO is very useful for LLMs. Its use in diffusion models is still under exploration. This paper studies how DPO can be used in the context of T2I diffusion models. The important consideration is that existing methods assign the same win/lose pair to all the intermediate steps. This is problematic. This paper thus generated two images at each step and compares the two images to obtain win/lose labels. The authors propose a way to compute such preference at later steps.

**Strengths:**

This paper's writing looks fine. The problems are presented in a right way.

**Weaknesses:**

- The proposed framework in Fig. 1(b) is almost the same as SPO (Liang et al., 2024). It is different in that SPO has a hyperparameter defining the number of images generated at each step, but this is very minor.

- Authors claim that this is the first framework that explicitly considers the properties of diffusion models for DPO. This is very wrong, because the same has been done in SPO.

- I understand that ArXiv papers may not be required to be cited, but it is important to acknowledge people's contribution in a proper way. It's very inappropriate to propose an identical method while claiming you are the first.

- The only difference, if I'm correct, is enhancing the diversity of noisy samples by increasing their reward gap. However, this is not claimed as the major contribution of this work. In fact, if this is the main contribution, this paper is not as problematic as now. The only critique would be incremental novelty etc.

- The comparison results of SPO look problematic too. There is almost no improvement of SPO over D3PO etc. This is very different from the report from SPO. Authors must carefully check the SPO paper and their open-source implementation.

**Questions:**

This paper should properly claim previous works.

**Details Of Ethics Concerns:**

Dear AC, SAC, and PCs,

This paper is problematic because the proposed method is almost identical to SPO. https://arxiv.org/abs/2406.04314

Although SPO is not published yet, it is not appropriate to say this paper is the first to taylor DPO to diffusion models. The authors use almost the same motivation, problem identification, and pipeline without clarifying that the same has already been fully introduced in SPO. Moreover, the reported results of SPO are not correct.

I'm strongly concerned about the academic integrity of this paper.

Best regards,
Reviewer

---

> ### Author Response · Authors · 2024-11-20
> **Response to Reviewer zqb2**
>
> Thank you for your review. Your overall reviews lie in the similarity between our method and SPO, as well as the doubts about the experimental results regarding SPO. We will now provide detailed responses to these questions.
>
> First, the similarity between our study and SPO (Liang et al., 2024) only lies in that both of us observe the inconsistency in the preference order between the intermediate-step outputs and final generations, and both of us compare the preference of intermediate-step outputs to address this inconsistency, although we **have different motivations and propose different solutions**. To this end, we have clearly and respectfully cited SPO (Liang et al., 2024) and discussed our differences in Lines 126-130 and Lines 204-208, and we have compared our method with SPO in experiments, as shown in Table 2, Table 3, and Figure 7. For example, we have mentioned that "SPO (Liang et al., 2024) also pointed out the problematic assumption ..." "Liang et al. (2024) demonstrated that ..." Therefore, we have never deliberately ignored the contribution of SPO.
>
> Second, beyond the above similarity, there are many differences between our study and SPO.
>
> - **The motivation is different.** Our study is motivated by the **theoretical discovery** of the mismatch between the trajectory-level ranking and step-level optimization. In comparison, SPO observed the inconsistency between intermediate-step outputs and final outputs from **visual demonstrations**. Specifically, we conduct detailed theoretical analysis and identify the following two issues of existing training framework: (1) *inaccurate preference order* and (2) *disturbed gradient direction*. These theoretical analysis motivates us to propose a new training framework of DPO tailored for diffusion models.
> - We use **totally different method to tackle the inaccurate preference order issue**. In SPO, the authors trained a step-wise reward model based on another assumption of the consistency between $x_t$ and images. In comparison, **we do not train a new model for reward evaluation**. Instead, we formulate the denoising     process as MDP and utilize the value function as the measurement of step-level reward.
> - We propose **different method and implementation to address the gradient issue**. First, we look back to the original formulation of DPO and ensure the same conditional input accordingly, which is very straightforward based on our theoretical analysis. Second, we notice that this operation potentially causes pairwise samples to be similar, so we introduce the gradient guidance of reward model in the training framework. This is one of the major contributions of this study, and it significantly improves the performance. In comparison, SPO chose to sample multiple outputs at each step for comparison.
>
> Therefore, we are the first to explicitly derive the theoretical flaws of previous DPO implementations in diffusion models based on distinct characteristics of diffusion models. Moreover, we are the first to leverage the gradient guidance technique of diffusion models in preference aligning to enhance the performance.
>
> Third, we obtain the results of SPO in Table 2 by running the officially released training code in [https://github.com/RockeyCoss/SPO](https://github.com/RockeyCoss/SPO/) using 45 animal-related prompts and evaluating the model on these prompts, following DDPO and D3PO for a fair comparison. The difference in prompts causes the difference in the reported results. We have introduced the detailed experimental settings in Lines 404-414 of the original manuscript and we will emphasis this difference for clarification.
>
> Furthermore, we also conduct a *new experiment* to compare the performance of TailorPO and SPO using the same prompts with SPO (Liang et al., 2024), *i.e.,* 4k prompts in the pick-a-pick dataset. We finetune SD-v1.5 using 4k prompts in the Pick-a-Pic training set, and evaluate the performance on 500 prompts in the Pick-a-Pic validation set. Results in the following table show that our methods still outperform SPO.
>
> |            | Aesthetic score | ImageReward |
> | ---------- | --------------- | ----------- |
> | SPO        | 5.887           | 0.1712      |
> | TailorPO   | 6.050           | 0.3820      |
> | TailorPO-G | 6.242           | 0.3791      |
>
> Besides the difference in prompts, the implementation of SPO in our study has slightly difference from (Liang et al., 2024). Due to the limit of resources, we finetuned SD-v1.5 with a small batch size of 2 on one V100 GPU for 10k samples, while Liang et al. (2024) finetuned SD-v1.5 with a large batch size of 40 on 4$\times$ A100 GPUs for 40k samples.

---

> ### Comment · Reviewer_zqb2 · 2024-11-21
> **Figure 1(b) is almost identical to Fig. 3 (c) in SPO (Liang et al. 2024)**
>
> Hi Authors,
>
> Thanks for your response. I would like to point out that Figure 1(b), which is a main contribution of the submission, is almost identical to Fig. 3 (c) in SPO (Liang et al. 2024).
>
> It is indicated the submission that "In contrast, we generate noisy samples from the same input xt and directly rank their preference order for optimization." This is the exactly the same as SPO.
>
> This paradigm has been proposed in SPO. Indeed you cited SPO, and you know SPO very well. But you didn't discuss how pipeline drawn in Fig. 3(c) differs from SPO (In fact there is no difference). To me, this difference has been intentionally ignored.
>
> I still hold strong reject and am skeptical about this paper's academic integrity.
>
> Regards,
> Reviewer

---

> > ### Author Response · Authors · 2024-11-21
> >
> > Thank you for your feedback. Your concern focuses on the similarity between the pipeline in our Figure 1 and SPO’s Figure 3(c). In fact, the only similarity is that both of us sample noisy outputs from the same $x_t$ to compare their quality. Nevertheless, we are motivated by a distinct theoretical discovery about the gradient issue in previous methods. Moreover, beyond this, there are many differences between our pipeline and SPO.
> >
> > First, our motivation for using the same $x_t$ is different. While SPO aims to make the quality comparison of paired images “reflects the denoising performance of this step alone,” we focus on the gradient issue in the optimization. Specifically, starting from the loss function of DPO and previous studies (D3PO), we notice their difference in the formulation of conditional inputs of the generative probability. Then, we consider how this setting affects the training. To this end, we derive the gradient of the previous loss function and identify that the gradient direction may be disturbed by different inputs $x_t$. Therefore, we consider the setting of different inputs to be problematic and it is straightforward to follow DPO to use the same $x_t$ for sampling and training. Furthermore, we also prove that using the same $x_t$ yields a correct and stable optimization direction.
> >
> > Second, beyond Figure 1, Figure 3 of our paper provides a more detailed introduction to our method, which also demonstrates the difference between our method and SPO. There are several major differences. (1) We compare the preference of intermediate-step outputs by directly estimating the step-wise reward, instead of training an additional reward model. (2) Given a pair of outputs from the same $x_t$, we consider that the similarity between them may affect the training effectiveness. Therefore, we propose to use the gradient of reward models to guide the generation process. In this way, we enlarge the difference between the two samples and boost the training effectiveness. (3) The output which is generated using the gradient guidance to achieve a high reward is utilized for sampling in the following steps.

---

### Official Review · Reviewer_nzop · 2024-11-03

**Soundness:** 2
**Presentation:** 3
**Contribution:** 2
**Rating:** 6
**Confidence:** 2

**Summary:**

A. The paradigm of formulating the backward diffusion process in reinforcement learning, as contributed by DDPO, Diffusion-DPO, and D3PO, is well explored.

B. In reinforcement learning, reducing fluctuations in advantage estimation by introducing value function is unanimously critical and well-explored in GAE and its related works. A more accurate estimation of advantages benefits reinforcement learning algorithms by applying policy optimization methods like PPO.

This work, TailorPO and TailorPO-G, is an instance of A + B among all possible formulations of similar ideas. The loss function used to train the diffusion model (policy) is modified by fixing the intermediate noisy sample $x_t$ to reduce fluctuations in estimating the effect of actions. Similar to GAE, $r_t$ serves as a value function that helps judge the quality of an action (referred to as the preference order in this paper). Backpropagating through the reward function $r$ helps identify the optimal $x_t$ (best/worst) as a data point for preference training.

**Strengths:**

* This paper claims to be the first to combine A and B, which I regard as a factual novelty under the formulations of D3PO and DDIM.
* The paper introduces the backpropagation of the reward function to generate data points for RL training, which may be useful for generating data points for other preference-learning algorithms.
* The efficacy of TailorPO(-G) is demonstrated with fewer than 10k samples.
* I believe the derivation of $\nabla_{\theta} \mathcal{L}$ is correct in Equation (11).  The neat form of Equation (11) is inspiring to me.
* Equation (12) provides insight into estimating reward expectations. I believe the property is correct and valuable for other applications if the parameters remain intact.
* I acknowledge that the paper is worth publishing, but it falls below the standard expected for ICLR (see below).

**Weaknesses:**

1. In Equation (12), the fact that $r_t$ can be approximated by $\epsilon_\theta$ **during training** is questionable. It is unclear whether $r_t$ can still be accurately estimated using either $\epsilon_\theta$ or $\epsilon_{\theta'}$ after the model parameter is updated from $\theta$ to $\theta'$, as prior work is proven under the initial $\theta$. This raises concerns about whether the optimization method remains effective after the parameter update, where $\theta'$ is significantly different from $\theta$. The absence of a theoretical argument hinders the soundness.
2. The aesthetic score using DDPO has reached its best quality (>8) with 40k reward queries. Based on point 1., I doubt whether it can practically achieve this score. The authors should demonstrate how their method can achieve the best quality of DDPO/D3PO on ANY of the rewards.
3. I assessed this work as an interpolation anchored from A and B; the novelty is, therefore, upper-bounded. Even if they reach the state of the art, predictable if the technique in B is well-implemented, they bring limited new knowledge to the community. Given the paper's novelty, I have set its upper bound for evaluation marginally above the acceptance threshold. Due to unresolved soundness issues, the paper stands slightly below the acceptance threshold.


Typos I found:
- Line 144.
- Line 312.

Factual Error:
1. Line 154: The DPO method is NOT first proposed to fine-tune large language models to align with human preferences unless the "preferences" refer to preference datasets. In this case, the authors should clarify.

**Questions:**

1. DDPO/D3PO-related methods cannot retain the Fokker-Planck equation of the original model, where probability flow ODE relies on this property. Can we use a trained policy to generate deterministically like probability flow ODE? Although challenging, this feature could add value if developed.
2. I would consider raising the rating (even above the upper bound) if the paper includes a more thorough formulation. For example, in reinforcement learning, the contribution to the value function is a relatively small extension of GAE (2015); I recommend incorporating at least a TD(λ) formulation. Additionally, as DDIM, DDPM, and other schedulers are subclasses of EDM, reformulating your framework under EDM could enable greater adaptability to alternative schedulers. Although EDM is not yet widely adopted in this community, such a formulation may enhance the paper’s applicability.
3. See Weaknesses

**Details Of Ethics Concerns:**

While the user study by issuing questionnaires appears to be low-risk and may be exempt from formal IRB review, the inclusion of a brief statement on ethical considerations or participant consent would enhance transparency.

---

> ### Author Response · Authors · 2024-11-20
> **Response to Reviewer nzop (part 1)**
>
> Thank you for your insightful comments. We will try our best to answer all your concerns. Please let us know if you still have further concerns, so that we can further update the response ASAP.
>
> **Q1: About the estimation of $r_t$ during training. "whether $r_t$ can still be accurately estimated using either $\epsilon_\theta$ or $\epsilon_{\theta'}$ after the model parameter is updated from $\theta$ to $\theta'$... whether the optimization method remains effective after the parameter update, where $\theta'$ is significantly different from $\theta$ ..."**
>
> **A1:** This is a good question. During training, $r_t$ can still be estimated by Eq. (12). This is because the approximation of $r_t(c, x_t) \triangleq \mathbb{E}[r(c,x_0)|c, x_t] \approx r(c, \hat{x}_0(x_t))$ is derived based on (1) the proof of $\mathbb{E}[x_0|c,x_t]=\hat{x}_0(x_t)$ and (2) the estimation of $\mathbb{E}[r(c,x_0)|c, x_t]\approx r(c, \mathbb{E}[x_0|c,x_t])$ in Proposition 1. Both proofs are not limited to a specific parameter $\theta$ and can be extended to $\theta'$ after training.
>
> First, the proof of $\mathbb{E}[x\_0|c,x\_t]=\hat{x}\_0(x\_t)$ is based on Tweedie’s formula, and the formula itself has no dependence on the model parameters. Specifically, Chung et al., (2023) provided the following proof in Appendix A of their paper. Given a noisy latent representation $x_t$, the conditional probability of $x_0$ can be written as $p(x\_0|x\_t)=p_0(x_t)\exp(x_0^TT(x_t)-\varphi(x_0))$, where $p\_0(x\_t)=\frac{1}{(2\pi (1-\bar{\alpha}\_t))^{d/2}} \exp(-\frac{\Vert x\_t \Vert^2}{2(1-\bar{\alpha}\_t)}), T(x\_t)=\frac{\sqrt{\bar{\alpha}\_t}}{1-\bar{\alpha}\_t}x\_t$, and $\varphi(x\_0)=\frac{\bar{\alpha}\_t \Vert x\_0 \Vert ^2}{2(1-\bar{\alpha}\_t)}$.
> According to this equation and Tweedie’s formula, we have $\mathbb{E}[x\_0|c,x\_t]=\frac{1}{\sqrt{\bar{\alpha}\_t}}(x\_t+(1-\bar{\alpha}\_t) \nabla\_{x\_t} \log p\_t (x\_t))$.
> To this end, both $\epsilon_{\theta}$ and $\epsilon_{\theta'}$ in the diffusion model represent an estimation for the term $\nabla_{x_t}p_t(x_t)$. Therefore, we can still estimate $\mathbb{E}[x_0|c,x_t]$ in the current model.
>
> Second, the estimation of $\mathbb{E}[r(c,x_0)|c, x_t]\approx r(c, \mathbb{E}[x_0|c,x_t])$ in Proposition 1 is proven based on the Jensen gap upper bound [1] given the reward function $r(\cdot)$, and it is agnostic to the model parameter of the diffusion model.
>
> [1] Gao et al., Bounds on the Jensen Gap, and Implications for Mean-Concentrated Distributions. arXiv:1712.05267.
>
> ---
>
> **Q2: About best quality. "... whether it can practically achieve this score. The authors should demonstrate how their method can achieve the best quality of DDPO/D3PO on ANY of the rewards."**
>
> **A2:** Thank you for the helpful suggestion. First, Figure 4 has shown that given the JPEG compressibility as the reward, out method can achieve the best quality of DDPO (>-10) within 4k paired samples, while DDPO needs more than 20k paired samples (according to Figure 4 of DDPO (Black et al., 2024)).
>
> Second, we have followed your suggestion to *conduct a new experiment*, where we take the aesthetic scorer as the reward model to finetune SD-v1.5 using DDPO, D3PO, and TailorPO on 40k paired samples. We report the change of the reward scorer during the training process in Figure 9 of the revised manuscript (the fine-tuning of D3PO is still in progress and we will update the result as soon as possible). We observe three phenomena from this figure.
>
> (1) TailorPO increases the aesthetic scorer the most effectively with less than 20k paired samples. This means that we can use fewer samples than other methods to achieve a good performance.
>
> (2) Although DDPO reaches the highest aesthetic score at 40k samples, we observe the severe reward hacking problem with the generated images. We provide some examples in Figure 9.  All these images are unnatural with the same color, same style, and similar background (yellow leaves). Therefore, instead of fine-tuning the diffusion model with too many samples to achieve an extremely high reward score, we would suggest controlling the number of samples to strike a balance between the good image quality and a high reward score.
>
> (3) D3PO is less effective than both DDPO and TailorPO, and this conclusion is consistent with Figure 3 of D3PO's original paper. This phenomenon also supports our discovery of its inherent issues about preference order and gradient direction.

---

> ### Author Response · Authors · 2024-11-20
> **Response to Reviewer nzop (part 2)**
>
> **Q3: About novelty and soundness.**
>
> **A3:** Thank you for the comment. We would like to discuss the novelty issue here and answer your concerns about soundness in the corresponding questions (Q1-Q2 and Q4-Q7).
>
> First, this study is not a simple A+B-styled work. Although we use a similar method with GAE [1] to introduce the value function for reward evaluation in aligning diffusion models, we are not simply combining these works. Instead, **we have a complete analysis framework tailored for aligning diffusion models**. Specifically, we start by rethinking the existing aligning framework of diffusion models and identifying the mismatch between the trajectory-level preference ranking and step-level optimization. This issue is substantiated by our theoretical analysis of the (1) inaccurate preference order and (2) disturbed gradient direction. Then, we propose methods to address these issues based on the theoretical analysis. Beyond introducing the value function for reward evaluation to address the inaccurate preference order, we revise the training framework of D3PO to align the gradient direction in optimization, and this is also a major contribution of this study. Moreover, we notice the potential impact of TailorPO on the similarity of paired samples, and novelly design TailorPO-G tailored for diffusion models to further improve the effectiveness. Finally, we conduct various experiments to demonstrate the effectiveness and generalization ability of our methods.
>
> Second, this study has a distinctive contribution to the community of generative models, especially the alignment of generative models. We identify potential issues in the existing DPO-styled aligning framework and provide a new framework tailored for the diffusion pipeline. Our framework can also be extended to various scenarios including aligning the generation of videos and 3D objects based on diffusion models.
>
> ---
>
> **Q4: Typos and the descriptions. "The DPO method is NOT first proposed to fine-tune large language models to align with human preferences unless the "preferences" refer to preference datasets."**
>
> **A4:** Thank you for your careful review. We have followed your suggestions to correct the typo and clarify the description of DPO in Line 155: "The DPO method is originally proposed to fine-tune large language models to align with human preferences based on paired datasets."
>
> ---
>
> **Q5: "Can we use a trained policy to generate deterministically like probability flow ODE?"**
>
> **A5:** Thank you, but I do not fully understand this question, so I would like to first confirm it with you. You mentioned that "DDPO/D3PO-related methods cannot retain the Fokker-Planck equation of the original model," and I am not sure what component of the FK equation is broken by DDPO/D3PO-related methods. Furthermore, in my understanding, although DDPO and D3PO change the model parameter, the model is still learned to approximate the score function $\nabla_x \log p_t(x)$ in the probability flow ODE, constrained by a KL regularization term. Therefore, I am confused about which operation prevents the model from deterministically generating. Could you kindly explain more about this concern? I hope we can discuss more on this question together during the rebuttal period, so as to help us understand and address your concern.

---

> ### Author Response · Authors · 2024-11-20
> **Response to Reviewer nzop (part 3)**
>
> **Q6: "would consider raising the rating (even above the upper bound) if the paper includes a more thorough formulation."**
>
> **A6:** Thank you for the insightful suggestion. We would like to provide a more thorough formulation of the value function and the diffusion framework. We have added the formulation of the value function in Section 3.2 of the revised manuscript. For the formulation of diffusion models based on EDM, due to the page limit and considering the readability of the paper, we add it to Appendix C of the revised manuscript.
>
> **Value function.** By formulating the denoising process of the diffusion model as MDP in Eq. (6), we aim to maximize the action value function at time $t$, *i.e.,* $Q(s,a)=\mathbb{E}[G_t|S_t=s,A_t=a]$, where $G_t$ represents the cumulative return at step $t$. We define  $G_t$ in the general form of $TD(\lambda)$, $G_t^{\lambda}=(1-\lambda)\sum_{n=1}^{T-t-1}\lambda^{n-1}G_t^{(n)}+\lambda^{T-t-1}G_t^{(T-t)}$, where $G_t^{(n)}=\sum_{i=1}^n \gamma^{i-1}R_{t+i}+\gamma^nV(S_{t+i})$ denotes the estimated return at step $t$ based on $n$ subsequent steps. Here, we simplify the analysis to $TD(1)$ and it degrades to Monte Carlo method. In other words, we have $G_t^\lambda=G_t^{(T-t)}=\sum_{i=1}^{T-t} \gamma^{i-1}R_{t+i}+\gamma^nV(S_{t+i})$. In the scenario of diffusion models, there is no intermediate feedback $R_t$ for intermediate steps and we assume $R_t=0$ for $t<T$. Therefore, the cumulative return can be further simplified as $\gamma^n V(S_{T})$. By setting $\gamma=1$ and $V(S_T)=R_T=r(c, x_0)$, which is the reward value of generated images, we have $Q(s,a)=\mathbb{E}[r(c,x_0)|S_t=(c,x_{T-t}, A_t=x_{T-t-1})]=\mathbb{E}[r(c,x_0)|c, x_{T-t-1}]$.
>
> **Diffusion framework**. Diffusion models contain a forward process and a reverse denoising process. Given an input $x_0$ sampled from the real distribution $p_\text{data}$, the forward process can be formulated as follows (EDM, [1]), which is a uniform formulation for DDPM, DDIM, and other methods.
> $$
> x_t=s_tx_0+s_t\sigma_t\epsilon
> $$
> where $x_t$ is the noisy sample at timestep $t$. $s\_t$ represents a scale schedule coefficient and $\sigma_t$ represents a noise schedule coefficient. At timestep $t$, we have $p(x\_t|x\_0)\sim\mathcal{N}(s\_t x\_0, s\_t^2\sigma\_t^2 I)$. $s\_t$ and $\sigma\_t$ are usually selected to ensure the final output $x\_T$ follows a certain Gaussian distribution.
>
> The reverse process aims to recover the distribution of original inputs $x\_0$ from a Gaussian noise $x\_T$. According to [1], the reverse ODE process is given as follows.
> $$
> dx=[\frac{\dot{s}\_t}{s\_t}x-s\_t^2\dot{\sigma}\_t\sigma\_t\nabla\_x\log p(\frac{x}{s\_t};\sigma\_t)]dt
> $$
> where $\dot{s\_t}$ and $\dot{\sigma}\_t$ denote the time derivative. $\nabla_x\log p(\frac{x}{s_t};\sigma_t)$ is the score function, which is usually approximated by a neural network, denoted by $s_\theta(\cdot)$. Replacing this term in the above equation, we can solve the reverse process. For a set of discrete timesteps, we can obtain a sequence $[x_T, x_{T-1}, \ldots, x_t, \ldots, x_0]$, and our study focuses on the optimization of $s_\theta(\cdot)$ at each timestep to generate $x_0$ with better image quality.
>
> Subsequently, the predicted $\hat{x}\_0$ at the step $t$ can be represented $\hat{x}\_0(x\_t)=\frac{1}{s\_t}(x\_t+s\_t^2\sigma\_t^2 s\_{\theta}(x\_t))$. Then, the step-wise reward value of $x\_t$ can be estimated based on $\hat{x}\_0(x\_t)$. Similarly, the conditional score function used in our gradient guidance can be rewritten as $\nabla_x\log p (\frac{x}{s_t}|r_\text{high};\sigma_t)=\nabla_x\log p (\frac{x}{s_t};\sigma_t)+\nabla_x\log p (r_\text{high}|\frac{x}{s_t};\sigma_t)$. The first term is estimated by the neural network $s_\theta(\cdot)$, and the second term can be approximated following Eq.(13) of our paper.
>
> [1] Karras et al., Elucidating the Design Space of Diffusion-Based Generative Models. NeurIPS 2022.

---

> ### Author Response · Authors · 2024-11-20
> **Response to Reviewer nzop (part 4)**
>
> **Q7: About user study. "the inclusion of a brief statement on ethical considerations or participant consent would enhance transparency."**
>
> **A7:** Thank you. We would like to provide more details about the user study, and we have followed your suggestion to add these details and discuss the ethical considerations in Appendix D of the revised manuscript.
>
> - Participant: We collect feedback from five annotators in the original manuscript, and we include five more annotators during the rebuttal phase. All annotators acknowledge that their efforts will be used to evaluate the performance of different methods in this paper.
>
> - Task instruction: The human annotators are given several triplets of ($c, x^{(a)}_1, x^{(b)}_0$), where $c$ is the text prompt and $x^{(a)}_1$ and $x^{(b)}_0$ represent the image generated by the model finetuned by method $a$ and method $b$, respectively. Then, the annotator is asked to compare two images from the perspective of alignment, aesthetics, and visual pleasantness. If both images in a pair look very similar or are both unappealing, then they should label “draw” for them. Otherwise, they label the image with the "win" and "lose" tags. In this way, for each pair of comparing methods, we have 225 triplets of ($c, x^{(a)}_1, x^{(b)}_0$) and each annotator label 225 "win/lose" or "draw" tags.
>
> - Mitigation: In order to avoid user bias, we hide the source of $x^{(a)}_1$ and $x^{(b)}_0$ and randomly place their order to annotators.

---

> ### Comment · Reviewer_nzop · 2024-11-22
> **Reply to rebuttal**
>
> **Context:**
> - Regardless of whether it is implemented in DDIM, DDPM, or other schedulers, given a dataset $\mathcal{D}$, a trained diffusion model $\epsilon_\theta$ is trained to fit $\nabla_{x_t}\log p_t(x_t)$ of the dataset under some rescheduling (SDE by Song et al., EDM by Karras et al.).
> - Theoretical assumption: I assume the base model $\epsilon_\theta$ perfectly implements $\nabla_{x_t}\log p_t(x_t)$.
> - The evolution of $p_t(x_t)$ can be derived from the Fokker-Planck equation with the forward SDE. The form is easier if formulated in EDM, i.e., as a mixture of Gaussians.
>
> I acknowledge the author's effort in the additional formulation of EDM and TD\($\lambda$). However, I was expecting the author to fundamentally formulate within this framework (starting from Equation 1). The current presentation in Appendix E appears to be a redundant add-on. Since I understand the required work is formidable, I will not decrease my rating without including them.
>
> My main concern remains an unsound **technical flaw**:
> 1. I believe that $r_t(c, x_t) \approx r(c, \hat{x}(x_t))$ if you implement $\hat{x}(x_t)$ using $\epsilon_\theta$.
> 2. I'm not convinced that you use $\epsilon_{\theta'}$ to implement $r_t(c, x_t) \approx r(c, \hat{x}(x_t))$. Specifically, "To this end, both $\epsilon_{\theta}$ and $\epsilon_{\theta'}$ in the diffusion model represent an estimation for the term $\nabla_{x_t}\log p_t(x_t)$" (I assume $\nabla_{x_t}p_t(x_t) \to \nabla_{x_t}\log p_t(x_t)$ is a typo). Once $\theta$ is modified to $\theta'$, it does **not** perfectly implement $\nabla_{x_t}\log p_t(x_t)$.
> 3. As explained by my theoretical argument, I've already predicted that this method will be effective **only at the beginning of the training**. Figure 9 further fortifies my confidence in this technical flaw.

---

> > ### Author Response · Authors · 2024-11-24
> >
> > Thank you for your feedback. We sincerely appreciate your suggestions on rigorous formulations. We have modified the formulation of the value function with $TD(\lambda)$, and added the formulation of EDM in the Appendix. We agree that formulation within the EDM will significantly improve the applicability of our method, and we plan to use this formulation in our future explorations of diffusion models. In this paper, considering that this study focuses on identifying potential flaws in existing works and then proposing a new training framework, we follow previous studies to use a simpler formulation (DDIM) in the main text for better comparison and readability.
> >
> > Regarding the concern about the estimation of the step-wise reward, we have conducted several new experiments to answer your concern. First, we compare the estimated value $r(c,\hat{x}\_0(x\_t))$ with  $r\_t(c,x\_t)\triangleq \mathbb{E}[r(c,x\_0)|c,x\_t]$ at different checkpoints to verify the reliability of using $\theta'$ after training for estimation. For the fine-tuned model $\epsilon_{\theta'}$, we sample 100 pairs of $(c,x\_t)$ at each timestep $t\in\\{12,8,4,1\\}$.  Give each pair of $(c,x\_t)$, we sample 100 images $x\_0$ based on $x\_t$, query reward values of all $x\_0$, and then compute $r\_t(c,x\_t)=\mathbb{E}[r(c,x\_0)|c,x\_t]$ as the ground truth of the step-wise reward. Then, we compute the estimated value $r(c,\hat{x}\_0(x\_t))$ based on the fine-tuned parameters $\theta'$. The following tables report the average relative error $\mathbb{E}[\vert\frac{r\_t(c,x\_t) - r(x, \hat{x}\_0(x\_t))}{r\_t(c,x\_t)}\vert]$ at different timesteps $t$ in different models (we use the aesthetic scorer and JPEG compressibility as the reward model, respectively).
> >
> > 👇Average relative error of aesthetic score.
> >
> > | timestep $t$                                       | 12     | 8      | 4      | 1      |
> > | -------------------------------------------------- | ------ | ------ | ------ | ------ |
> > | Pre-trained model $\epsilon_\theta$                    | 0.0545 | 0.0378 | 0.0132 | 0.0047 |
> > | $\epsilon_{\theta'}$ after training on 10k samples | 0.0353 | 0.0176 | 0.0106 | 0.0033 |
> > | $\epsilon_{\theta'}$ after training on 40k samples | 0.1330 | 0.0283 | 0.0132 | 0.0070 |
> >
> > 👇Average relative error of JPEG compressibility
> >
> > | timestep $t$                                       | 12     | 8      | 4      | 1      |
> > | -------------------------------------------------- | ------ | ------ | ------ | ------ |
> > | Pre-trained model $\epsilon_\theta$                    | 0.2263 | 0.1259 | 0.0390 | 0.0070 |
> > | $\epsilon_{\theta'}$ after training on 10k samples | 0.2492 | 0.1440 | 0.0425 | 0.0074 |
> > | $\epsilon_{\theta'}$ after training on 40k samples | 0.1566 | 0.0341 | 0.0113 | 0.0066 |
> >
> > These results demonstrate that after fine-tuning, the model $\epsilon_{\theta'}$ achieves a small error as the pre-trained model $\epsilon_\theta$ does. Moreover, our DPO-based loss function **does not require an accurate reward value, but only needs the preference order of samples**. Even if there is a small estimation error for the step-wise reward, it does not affect the preference order between paired samples, thus having little effect on training. Therefore, the modified parameter $\theta'$ can still be utilized to reliably estimate the step-wise reward.
> >
> > Second, we would like to discuss the reason for using the fine-tuned parameter $\theta'$ instead of the pre-trained parameter $\theta$.  In our scenario, we aim to estimate the value function of $x_t$ in the current model during training, and this requires the expectation of images *generated by the current model* $\epsilon_{\theta'}$ given $x_t$, *i.e.,* $\mathbb{E}[x_t|c,x_t,\theta']$. To this end, the pre-trained parameter $\theta$ yields the expectation of images generated by the pre-trained model, $\mathbb{E}[x_t|c,x_t,\theta]$, rather than the current model. In other words, the pre-trained parameter $\theta$ can only estimate the value function of $x_t$ in the pre-trained model, but cannot be used in models after training. Therefore, we choose to use the fine-tuned parameter $\theta'$. Considering the proof based on Tweedie's formula in our previous response, this can be considered that we use a shifted approximation for $\nabla_{x_t}\log p_t(x_t)$ to estimate images $x_0$ in a shifted distribution with high reward values.
> >
> > Third, we are conducting new experiments to investigate the performance of our methods on more reward models and base models, after training on more than 10k samples. This experiment takes more time and we will update the result as soon as possible.

---

> > > ### Comment · Reviewer_nzop · 2024-11-25
> > > **Reply**
> > >
> > > ## Reply to the Soundness
> > >
> > > ### On Empirical Results
> > > Given the importance of empirical results in this domain, I can overlook theoretical unsoundness if supported by **substantial** experimental evidence.
> > > The goal of the arguments is to refute mine by providing empirical results rather than characterizing $\nabla_x \log p_t(x_t)$ after training $\theta$. Writing a consistent story with theoretical and empirical results would be better.
> > >
> > >
> > > I still question the empirical results:
> > > 1. Error bars are missing from both tables, as commented on November 25, 2024, at 01:15.
> > > 2. What is the maximum value of $t$? The author should include this value in the table.
> > >
> > > ### On Theoretical Results
> > > Having fixed condition $c$, let $\mathcal{D}'$ denote the image distribution generated by $x_0 | c, x_t, \theta'$, obtained from your trained model $\theta'$. Let $\epsilon_\phi$ be the unique diffusion model that perfectly fits $\mathcal{D}'$. Your argument introduces the hypothesis (H) that $\epsilon_\phi \equiv \epsilon_{\theta'}$, with the rest following from Tweedie's formula. We should question the validity of H.
> > >
> > > My intuition, supported by Figure 9, suggests that H is **false**, as $\epsilon_\phi$ is constrained by the Fokker-Planck equation, while $\epsilon_{\theta'}$ is not. Therefore, I cannot accept this method on a theoretical basis.
> > >
> > > For better understanding, I'd like to ask if you can briefly explain Tweedie's formula **in the context of the paper**, including the specific conditions under which the formula applies.
> > >
> > > ## Other Concerns
> > >
> > > I raised my rating to 6 but lowered my confidence to 2; here's why:
> > >
> > > Having read Reviewer zqb2's comments and the paper SPO, I cannot ignore the similarities between the components. Given SPO's contributions, I perceive the remaining contributions on the table to be:
> > > 1. Theoretical contributions
> > > 2. TailorPO-G
> > >
> > > I have already expressed my concerns about the theoretical contributions. If I were to assign a score of 6 (marginally above the acceptance threshold), it would be solely due to TailorPO-G. However, considering the contributions of TailorPO-G, **it still falls short of my expectations for ICLR papers**. This score is **conditional** on the resolution of the integrity issue raised by zqb2.

---

> > > > ### Author Response · Authors · 2024-11-28
> > > > **Response to follow-up comments (part 1)**
> > > >
> > > > Thank you for your kind feedback and engagement in the discussion. Below, we provide answers to all your questions.
> > > >
> > > > **1. About concerns on empirical results.**
> > > >
> > > > Q: "Error bars are missing from both tables.""What is the maximum value of $t$?"
> > > >
> > > > A: Thank you. We have followed your suggestion to compute and report the standard deviation in the following tables.
> > > >
> > > > 👇Average relative error of aesthetic score.
> > > >
> > > > | timestep $t$                                       | 12                 | 8                  | 4                  | 1                  |
> > > > | -------------------------------------------------- | ------------------ | ------------------ | ------------------ | ------------------ |
> > > > | Pre-trained model $\epsilon_\theta$                | 0.0545 $\pm$0.0427 | 0.0378 $\pm$0.0287 | 0.0132 $\pm$0.0089 | 0.0047 $\pm$0.0051 |
> > > > | $\epsilon_{\theta'}$ after training on 10k samples | 0.0353 $\pm$0.0345 | 0.0176 $\pm$0.0160 | 0.0106 $\pm$0.0080 | 0.0033 $\pm$0.0029 |
> > > > | $\epsilon_{\theta'}$ after training on 40k samples | 0.1330 $\pm$0.0320 | 0.0283 $\pm$0.0231 | 0.0132 $\pm$0.0084 | 0.0070 $\pm$0.0047 |
> > > >
> > > > 👇Average relative error of JPEG compressibility
> > > >
> > > > | timestep $t$                                       | 12                 | 8                  | 4                  | 1                  |
> > > > | -------------------------------------------------- | ------------------ | ------------------ | ------------------ | ------------------ |
> > > > | Pre-trained model $\epsilon_\theta$                | 0.2263 $\pm$0.0524 | 0.1259 $\pm$0.0333 | 0.0390 $\pm$0.0101 | 0.0070 $\pm$0.0039 |
> > > > | $\epsilon_{\theta'}$ after training on 10k samples | 0.2492 $\pm$0.0390 | 0.1440 $\pm$0.0279 | 0.0425 $\pm$0.0071 | 0.0074 $\pm$0.0016 |
> > > > | $\epsilon_{\theta'}$ after training on 40k samples | 0.1566 $\pm$0.0925 | 0.0341 $\pm$0.0221 | 0.0113 $\pm$0.0077 | 0.0066 $\pm$0.0016 |
> > > >
> > > > The maximum value of $t$ is the number of denoising steps $T=20$.
> > > >
> > > > ---
> > > >
> > > > Besides the above experiment, we have also conducted new experiments on more different reward models and base models to demonstrate that the effectiveness of TarilorPO is not limited to early iterations. First, we fine-tune SD-v1.5 using DDPO, D3PO, and TailorPO to improve the JPEG compressibility, respectively. Figure 10 (left) shows that all methods converge in later epochs, and TailorPO exhibits the **fastest** learning efficiency and achieves the **highest reward value**. Furthermore, we also fine-tune SD-v2.1-base using DDPO and TailorPO by taking the aesthetic scorer as the reward model (other settings including learning rate are the same as Section 4). Figure 10 (right) shows that TailorPO significantly outperforms DDPO after training on 40k samples, and it is effective throughout the learning process.
> > > >
> > > > Regarding the phenomenon in Figure 9, the learning of TailorPO is a bit slower after 30k samples because of the constraint in the implementation of the loss function. Following D3PO (Yang et al., 2024a), we constrain values of $\frac{\pi_\theta(x^w_{t-1}|x_t,c)}{\pi_\text{ref}(x^w_{t-1}|x_t,c)}$ and $\frac{\pi_\theta(x^w_{t-1}|x_t,c)}{\pi_\text{ref}(x^w_{t-1}|x_t,c)}$ to be within the range of $[1-\delta, 1+\delta]$.  This constraint helps avoid the model being led too far away from the reference model. During the training process, the probability in the reference model ($\pi_\text{ref}(x^w_{t-1}|x_t,c)$ and $\pi_\text{ref}(x^l_{t-1}|x_t,c)$) keeps decreasing because both $x^w_{t-1}$ and $x^l_{x-1}$ are sampled from the fine-tuned model. Therefore, values of $\frac{\pi_\theta(x^w_{t-1}|x_t,c)}{\pi_\text{ref}(\cdot|x_t,c)}$ and $\frac{\pi_\theta(x^l_{t-1}|x_t,c)}{\pi_\text{ref}(\cdot|x_t,c)}$ increase during the training process. Upon their values reach the constraint, they are clipped into the range of $[1-\delta,1+\delta]$. In this case, the optimization on this pair of samples is restricted. Therefore, in late training process, there could be more samples reaching the constraint and the optimization probably gets slower. We will clarify this setting in our revised manuscript.

---

> > > > ### Author Response · Authors · 2024-11-28
> > > > **Response to follow-up comments (part 2)**
> > > >
> > > > **2. About concerns on theoretical results.**
> > > >
> > > > Thank you for the insightful comment. We would like to  first demonstrate the formulation and conditions of the Tweedie's formula. Then, we discuss how to apply Tweedie's formula in diffusion models. We will add these discussions in our paper for better understanding.
> > > >
> > > > *Tweedie's formula.* Let $p(y|\eta)$ denote any distribution that belongs to the *exponential family of probability distributions*, which is defined as those distributions whose density can be written as the following form.
> > > > $$
> > > > p(y|\eta)=p_0(y)\exp(\eta^T T(y)-\varphi(\eta))
> > > > $$
> > > > where $\eta$ is a canonical parameter of the family, $T(y)$ is a function of $y$, $\varphi(\eta)$ is the cumulant generating function which makes $p(y|\eta)$ integrate to 1, and $p_0(y)$ is the density up to a scale factor when $\eta=0$. Notably, the Gaussian distribution is a typical class of the above exponential family distribution. Then, the posterior mean $\hat\eta=\mathbb{E}[\eta|y]$ should satisfy $(\nabla_y T(y))^T\hat\eta=\nabla_y \log p(y)-\nabla_y \log p_0(y)$. Please refer to [1,2] for more details.
> > > >
> > > > *Application in diffusion models.* Given a distribution of natural images $D$, let $x_0$ denote the image in $D$, and $x_t=\sqrt{\bar{\alpha}_t}x_0+\sqrt{1-\bar{\alpha}_t}\epsilon$ denote the corresponding noisy sample. $p_t(x_t)$ denotes the distribution of $x_t$ obtained on images $x_0\sim D$. The conditional distribution of $x_t$ given $x_0$ is a Gaussian distribution, belonging to the *exponential family distribution*, and it can be written as follows.
> > > > $$
> > > > \begin{array}{ll}
> > > > p(x_t|x_0)&=\frac{1}{(2\pi (1-\bar{\alpha}_t))^{d/2}} \exp(-\frac{\Vert x_t - \sqrt{\bar{\alpha}_t}x_0 \Vert^2}{2(1-\bar{\alpha}_t)})\\ &=p_0(x_t)\exp(x_0^TT(x_t)-\varphi(x_0))
> > > > \end{array}
> > > > $$
> > > > where $p_0(x_t)=\frac{1}{(2\pi (1-\bar{\alpha}_t))^{d/2}} \exp(-\frac{\Vert x_t \Vert^2}{2(1-\bar{\alpha}_t)}), T(x_t)=\frac{\sqrt{\bar{\alpha}_t}}{1-\bar{\alpha}_t}x_t$, and $\varphi(x_0)=\frac{\bar{\alpha}_t \Vert x_0 \Vert ^2}{2(1-\bar{\alpha}_t)}$. According to Tweedie's formula, we have $\frac{\sqrt{\bar{\alpha}\_t}}{1-\bar{\alpha}\_t}\hat{x}\_0=\nabla\_{x\_t}\log p\_t(x\_t) + \frac{1}{1-\bar{\alpha}\_t}x\_t$. This equation can be rewritten as $\hat{x}\_0=\frac{1}{\sqrt{\bar{\alpha}\_t}}(x\_t+(1-\bar{\alpha}\_t)\nabla\_{x\_t}\log p\_t(x\_t))$, and the pre-trained model $\epsilon\_\theta$ approximates the $\nabla\_{x\_t}\log p\_t(x\_t)$ term.
> > > >
> > > > The aligning of the diffusion model can be considered from a similar perspective. Given another distribution $D'$ of natural images that *have high reward values*, we are actually fitting a new distribution $p'_t(x'_t)$ where $x'_t$ is obtained on images $x'_0\sim D'$. Similar to the above equation, the conditional distribution of $x'_t$ given $x'_0$ is also a Gaussian distribution. Therefore, based on Tweedie's formula, the posterior mean of $x'_0$ can be derived as $\hat{x}'\_0=\frac{1}{\sqrt{\bar{\alpha}\_t}}(x'\_t+(1-\bar{\alpha}\_t)\nabla\_{x'\_t}\log p'\_t(x'\_t))$. Here, we assume that the $\nabla\_{x'\_t}\log p'\_t(x'\_t)$ term can be approximated by the model $\epsilon\_{\theta'}$ after training, and the estimation based on $\epsilon\_\theta'$ has been verified by experiments in our last response. We will clarify all these analyses in the paper.
> > > >
> > > >
> > > > [1] Kim and Ye. Noise2Score: Tweedie’s Approach to Self-Supervised Image Denoising without Clean Images. NeurIPS 2021.
> > > >
> > > > [2] Chung et al. Diffusion Posterior Sampling for General Noisy Inverse Problems. ICLR 2023.

---

> > > > ### Author Response · Authors · 2024-11-28
> > > > **Response to follow-up comments (part 3)**
> > > >
> > > > **3. About other concerns.**
> > > >
> > > > We appreciate your efforts and acknowledge that our work shares certain similarities with SPO. However, we would like to clarify that our research was conducted entirely independently.  Although we have cited, discussed, and empirically compared with SPO in our paper, we would like to further emphasize the distinct contribution of our paper, which lies in the following perspectives.
> > > >
> > > > First, we discover distinct **theoretical findings** to support the design of our training framework, while SPO is mainly based on empirical intuitions.
> > > >
> > > > - We formulate the reward of each denoising step as the action value function in MDP, and **derive its theoretical formulation** $\mathbb{E}[r(c, x_0)|c, x_t]$ (Section 3.2).
> > > > - We **discover and theoretically prove the gradient issue** caused by previous methods for the first time (Section 3.2). Such analysis inspires us to sample from the same $x_t$, and we theoretically prove that this simple operation could address the gradient issue and align the optimization direction with preferences (Section 3.3).
> > > >
> > > > Second, we have the following **technical contributions**, which significantly differ from SPO.
> > > >
> > > > - We **directly evaluate** the preference quality of noisy samples based on the estimation for the action value function (Section 3.3), instead of training a new reward model based on uncertified assumptions.
> > > > - We **incorporate the gradient guidance** of reward models to enlarge the gap between paired samples to boost the aligning effectiveness for the first time. We also theoretically prove that this guidance pushes the model optimization towards high reward values from the perspective of gradient (Section 3.4 and Appendix B).
> > > >
> > > > Finally, experimental results demonstrate that TailorPO achieved **better performance and generalization ability than previous methods** including SPO (Section 4.1 and Section 4.2). This is because the estimation in Eq. (12) for the step-wise reward provides us with a more accurate and reliable preference label.  Furthermore, TailorPO-G further improved the aligning effectiveness.

---

### Official Review · Reviewer_zd3f · 2024-11-04

**Soundness:** 2
**Presentation:** 2
**Contribution:** 2
**Rating:** 6
**Confidence:** 3

**Summary:**

The paper presents a novel framework, TailorPO, aimed at aligning diffusion models with human preferences by directly optimizing for preference at each denoising step. This approach addresses a significant issue in existing direct preference optimization (DPO) methods, which often assume consistency between preferences for intermediate noisy samples and final generated images. The authors argue convincingly that this assumption can disrupt the optimization process, and their proposed solution is both innovative and timely, given the growing interest in controllable generative models.

**Strengths:**

1. Originality:
The paper introduces TailorPO, a novel framework that addresses a specific limitation in the application of direct preference optimization (DPO) to diffusion models. This represents a creative advancement in the field of generative modeling. The approach of generating noisy samples from the same input at each denoising step is innovative and directly addresses the issue of inconsistent preference ordering in intermediate steps of image generation.

2. Quality:
The paper is technically rigorous, providing a detailed theoretical analysis of the issues with existing DPO methods and a comprehensive empirical evaluation of TailorPO. The experiments are well-designed and demonstrate significant improvements over other methods in generating aesthetically pleasing images that align with human preferences.

3. Clarity:
The paper is well-organized, with a clear structure that logically progresses from problem formulation to solution proposal, followed by experimental validation.

4. Significance:
The work has high significance as it tackles a critical challenge in aligning generative models with human preferences, which is essential for the practical application of these models. The proposed TailorPO framework has the potential to influence future research in controllable generative modeling.

**Weaknesses:**

1. The user study, while valuable, is limited in scope with a small number of participants. A larger and more diverse user base would provide more robust evidence of the framework's effectiveness.

2. While the paper compares TailorPO with other DPO-style methods, it would be strengthened by including comparisons with the current state-of-the-art methods.

3. The paper could provide a more detailed discussion on the limitations of TailorPO, such as specific types of images or preferences that the model may struggle with, or scenarios where the framework might not be applicable.

**Questions:**

1. How does TailorPO perform compared to the current state-of-the-art methods in generative modeling, particularly in terms of image quality and alignment with human preferences? Benchmarking against state-of-the-art methods can provide a clearer picture of TailorPO's performance and its potential advantages.

2. Are there specific scenarios or types of images where TailorPO underperforms or fails to align with human preferences? If so, could the authors discuss these limitations and potential avenues for improvement?

---

> ### Author Response · Authors · 2024-11-20
> **Response to Reviewer zd3f (part 1)**
>
> Thank you for your valuable comments. We will try our best to answer all your concerns. Please let us know if you still have further concerns, so that we can further update the response ASAP.
>
> **Q1: About user study. "A larger and more diverse user base would provide more robust evidence of the framework's effectiveness."**
>
> **A1:** Thank you for the valuable comments. We have followed your suggestion to *conduct a broader user study* by including more users. Beyond five users included in our original manuscript, we ask another five users to compare the preference of images generated by different methods. Following the settings in Lines 469-475 and Appendix D, we collect feedback from five more users and report the results obtained from a total of 10 users as follows.
>
> |                          | TailorPO vs. DDPO | TailorPO vs. SPO | TailorPO-G vs. DDPO | TailorPO vs. SPO |
> | ------------------------ | ----------------- | ---------------- | ------------------- | ---------------- |
> | TailorPO/TailorPO-G win  | 59.33             | 54.22            | 59.24               | 53.90            |
> | Draw                     | 22.22             | 27.56            | 25.17               | 28.95            |
> | TailorPO/TailorPO-G lose | 18.45             | 18.22            | 15.59               | 17.15            |
>
> The result shows that TailorPO and TailorPO-G better align the model generations with human preference. We have accordingly revised the description and the result of the user study in Figure 7 in the revised manuscript.
>
> ---
>
> **Q2: About comparison with SOTA methods. "While the paper compares TailorPO with other DPO-style methods, it would be strengthened by including comparisons with the current state-of-the-art methods." "How does TailorPO perform compared to the current state-of-the-art methods ..."**
>
> **A2:** Thank you. **First**, this study aims to align a base generative model with human preference via fine-tuning, instead of training a new base model, so we compared our methods with existing fine-tuning methods. **Second**, we have actually included both PPO-style method (DDPO) and DPO-style methods (D3PO and SPO) for comparison. These methods are all proposed in 2023 and 2024, and they represent the current state-of-the-art methods in this direction. Table 2 and Figure 3 have shown that our methods achieved higher reward values than these methods.
>
> **Third**, we have followed your suggestion to *conduct a new experiment* to compare TailorPO with the state-of-the-art offline method, Diffusion-DPO (Wallace et al., 2024). Diffusion-DPO finetuned SD-v1.5 on the Pick-a-Pic dataset in an offline manner. Therefore, we also finetune SD-v1.5 using TailorPO on prompts in the Pick-a-Pic training set, taking the aesthetic scorer and ImageReward as the reward model, respectively. Then, we evaluate the performance using 500 prompts in the Pick-a-Pic validation set, as done by (Liang et al., 2024). The results of Diffusion-DPO in the following table are from (Liang et al., 2024).
>
> |               | Aesthetic score | ImageReward |
> | ------------- | --------------- | ----------- |
> | Diffusion-DPO | 5.505           | 0.1115      |
> | TailorPO      | 6.050           | 0.3820      |
> | TailorPO-G    | 6.242           | 0.3791      |
>
> The above table shows that our methods achieve higher reward values than Diffusion-DPO in both aesthetic score and ImageReward score. We have added this experiment in Appendix E.1 of the revised manuscript.

---

> ### Author Response · Authors · 2024-11-20
> **Response to Reviewer zd3f (part 2)**
>
> **Q3: About limitations. "could provide a more detailed discussion on the limitations of TailorPO," "could the authors discuss these limitations and potential avenues for improvement?"**
>
> **A3:** Like other methods based on an explicit pre-trained reward model, including DDPO, D3PO, and SPO, TailorPO has the potential of being prone to reward hacking [1], if we fine-tune the model on very simple prompts for too many iterations. It means that the generative model is overoptimized to improve the score of the reward model but fails to maintain the original output distribution of natural images. We provide some examples in Figure 12 of the revised manuscript to demonstrate this phenomenon. For example,  when we take JPEG compressibility as the reward, DDPO, D3PO, and our methods all generate images with a blank background.
>
> The problem of reward hacking is related to the quality of reward models. Given the fact that these pre-trained reward models are usually trained on a finite training set, they cannot *perfectly* fit the human preference for natural and visually pleasing images. Therefore, the optimization of generative models towards these reward models may lead to an unnatural distribution of images.
>
> In order to alleviate the reward hacking problem, TailorPO can be further improved from the following perspectives.
>
> - Using a better reward model that well captures the distribution of natural and visually pleasing images. A better reward model can avoid guiding model optimization towards unnatural images.
> - Utilizing the ensemble of multiple reward models to alleviate the bias of a single reward model. While each single reward model has its own preference bias, considering multiple reward models altogether may be able to alleviate the risk of falling into a single model. To this end, [2] has shown that the reward model ensembles can effectively address the reward hacking in RLHF-based fine-tuning of language models. Therefore, we are hopeful that the reward model ensembles are also effective for diffusion models.
> - Searching for a better setting of the hyperparameter $\beta$ in the loss function to strike a balance between natural images and high reward scores. In DPO-style methods, the coefficient $\beta$ controls the deviation from the original generative distribution (the KL regularization). In this way, we can search for a better value of $\beta$ to avoid the model being fine-tuned far away from the original base model. For example, [3] has provided a method to dynamically adjust the value of $\beta$.
>
> We have followed your suggestion to add all these discussions in Appendix G of the revised manuscript.
>
> [1] Skalse et al., Defining and Characterizing Reward Hacking. NeurIPS 2022.
>
> [2] Coste et al., Reward Model Ensembles Help Mitigate Overoptimization. ICLR 2024.
>
> [3] Wu et al., $\beta$-DPO: Direct Preference Optimization with Dynamic $\beta$. NeurIPS 2024.

---

> ### Author Response · Authors · 2024-11-28
>
> Dear Reviewer,
>
> We sincerely appreciate your efforts and insightful comments, which has helped us to improve our paper. We have followed your suggestions to conduct new experiments and to discuss the limitations in our response. We would like to confirm whether our response has addressed all your concerns. Please let us know if you have any further questions, and we will respond as soon as possible.
>
> Best regards,
> Authors

---

> > ### Comment · Reviewer_zd3f · 2024-12-03
> >
> > The authors have addressed my previous concerns. I would like adjust my scores. But based on reviewer zqb2's comments, I still have some concerns for the similarities of the main ideas between this work and SPO.

---

> > > ### Author Response · Authors · 2024-12-03
> > >
> > > Thank you for taking the time to review our response and raising your score. We sincerely appreciate your insightful suggestions, and we are glad to hear your previous concerns are addressed.
> > >
> > > Regarding the concern for the similarity between our work and SPO, we would like to clarify our similarities and differences. The similarity between our study and SPO only lies in that both of us evaluate the reward of each denoising step given the same input. To this end, we have cited, discussed, and empirically compared with SPO in our paper. Beyond this similarity, we would like to further emphasize the distinct contribution of our paper, which lies in the following perspectives.
> > >
> > > First, we discover distinct **theoretical findings** to support the design of our training framework, while SPO is mainly based on empirical intuitions.
> > >
> > > - We formulate the reward of each denoising step as the action value function in MDP, and **derive its theoretical formulation** $\mathbb{E}[r(c, x_0)|c, x_t]$ (Section 3.2).
> > > - We **discover and theoretically prove the gradient issue** caused by previous methods for the first time (Section 3.2). Such analysis inspires us to sample from the same $x_t$, and we theoretically prove that this simple operation could address the gradient issue and align the optimization direction with preferences (Section 3.3).
> > >
> > > Second, we have the following **technical contributions**, which significantly differ from SPO.
> > >
> > > - We **directly evaluate** the preference quality of noisy samples based on the estimation for the action value function (Section 3.3), instead of training a new reward model based on uncertified assumptions.
> > > - We **incorporate the gradient guidance** of reward models to enlarge the gap between paired samples to boost the aligning effectiveness for the first time. We also theoretically prove that this guidance pushes the model optimization towards high reward values from the perspective of gradient (Section 3.4 and Appendix B).
> > >
> > > Finally, experimental results demonstrate that TailorPO achieved **better performance and generalization ability than previous methods** including SPO (Section 4.1 and Section 4.2). This is because the estimation in Eq. (12) for the step-wise reward provides us with a more accurate and reliable preference label.  Furthermore, TailorPO-G further improved the aligning effectiveness.
> > >
> > > Thank you again for your reply and we hope this could answer your further concerns.

---

### Meta-Review · Area_Chair_pWm6 · 2024-12-26

**Metareview:**

The paper introduces the Tailored Preference Optimization (TailorPO) framework designed to align diffusion models with human preferences by optimizing for preference at each denoising step. This method tackles a key limitation (i.e., assuming the final generated images and intermediate noisy samples sharing consistent preference labels.) in existing direct preference optimization (DPO) techniques by ranking the intermediate samples based on their step-wise reward. The work provides the theoretical justification, and the proposed method TailorPO and TailorPO-G achieves better results than previous state-of-the-art. Most reviewers acknowledge the theoretical contributions. They also recognize the novelty and the superior performance of the proposed TailorPO-G. However, the reviewer  zqb2 raises the issue of academic integrity for the resemblance of some materials of the paper with SPO. In addition, other reviewers  request more experiments on baseline comparisons and further analyses (zd3f, nzop, zqb2, F2pk, ZmZ1), more discussions about the pros and cons of the proposed TailorPO (zd3f), and elaboration on the difference between TailorPO and TailorPO-G (F2pk). During the discussions, the authors provide detailed responses and address most of the concerns of (zd3f, nzop, F2pk, ZmZ1), except zqb2. The paper receives 4 borderline accept and 1 strong reject, leading to 5.0 on average.

However, other reviewers also acknowledge that there exists a strong correlation between SPO and the proposed method. Although the authors did add some paragraphs to describe the difference from SPO, I agree with reviewer zqb2 that the claims of the revised paper about its contributions in the abstract, introduction, and other parts of the paper remain a cause of confusion and need be further revised to acknowledge SPO more clearly and properly at the right places. For example, in the abstract Ln 23 to 24, "we are the first to consider the distinct structure of diffusion models and leverage the gradient guidance in preference aligning to enhance the optimization effectiveness." The first half may cause some confusion and should be revised by adding more context to better reflect the contributions of the paper while properly acknowledging SPO. Similarly, the authors should also properly polish the other parts of the paper to more specifically to reflect this, such as Ln 98 in the introduction, etc. Additionally, the claims of contributions should focus on the theoretical part and TailorPO-G.

Considering the current status of the manuscript, we do not think it can be accepted to ICLR'2025. However, we do encourage the authors to revise the manuscript thoroughly and address the aforementioned concerns, such as describing the correlation between Fig 1 and Fig 3 of SPO, and resubmit the paper to a future venue.

**Additional Comments On Reviewer Discussion:**

During the discussion period, the authors provided detailed responses to address most of the concerns raised by reviewers zd3f, nzop, F2pk, and ZmZ1. Their responses included further explanations, additional experimental results, and in-depth analyses of the proposed TailorPO and TailorPO-G methods. As a result, these reviewers increased their ratings from 5 to 6. Although the authors added new paragraphs and sentences elaborating on the differences between their method and SPO, some concerns raised by reviewer zqb2 remain. I agree with zqb2 that the paper’s claims regarding its contributions—particularly in the abstract, introduction, and other sections—could lead to confusion. These parts should be revised to acknowledge SPO clearly and properly before the paper gets published

---

### Decision · Program_Chairs · 2025-01-22

Reject